# SISALv3: A global speleothem stable isotope and trace element database

Nikita Kaushal[1*], Franziska A. Lechleitner[2*], Micah Wilhelm[3], Khalil Azennoud[4], Janica C. Bühler[5], Kerstin Braun[6], Yassine Ait Brahim[4], Andy Baker[7], Yuval Burstyn[8], Laia Comas-Bru[9], Jens Fohlmeister[10], Yonaton Goldsmith[11], Sandy P. Harrison[12], István G. Hatvani[13,14], Kira Rehfeld[5], Magdalena Ritzau[5], Vanessa Skiba[15, 16], Heather M. Stoll[17], József G. Szűcs[18], Péter Tanos[18], Pauline C. Treble[7,19], Vitor Azevedo[20], Jonathan L. Baker[21,22], Andrea Borsato[23], Sakonvan Chawchai[24], Andrea Columbu[25], Laura Endres[17], Jun Hu[26], Zoltán Kern[13,14], Alena Kimbrough[27], Koray Koç[28,29], Monika Markowska[30,31], Belen Martrat[32], Syed Masood Ahmad[33], Carole Nehme[34], Valdir Felipe Novello[5], Carlos Pérez-Mejías[21], Jiaoyang Ruan[35,36], Natasha Sekhon[37,38], Nitesh Sinha[35,36], Carol V. Tadros[7,19], Benjamin H. Tiger[39,40], Sophie Warken[41,42], Annabel Wolf[43], Haiwei Zhang[21] and SISAL Working Group members[+]

[1]Exeter College, University of Oxford, Oxford OX1 3DP, UK; Now at the American Museum of Natural History, New York 10024, United States of America
[2]Department of Chemistry, Biochemistry and Pharmaceutical Sciences, and Oeschger Centre for Climate Change Research, University of Bern, Freiestrasse 3, 3012 Bern, Switzerland
[3]Swiss Federal Institute for Forest, Snow and Landscape Research WSL, Zürcherstrasse 111, 8903 Birmensdorf, Switzerland
[4]International Water Research Institute, Mohammed VI Polytechnic University, Lot 660 Hay Moulay Rachid, 43150, Benguerir, Morocco
[5]Department of Geoscience and Department of Physics, Geo- und Umweltforschungszentrum (GUZ), Schnarrenbergstr. 94/96, 72076 Tübingen, Germany
[6]Arizona State University, Institute of Human Origins, PO Box 878404, Tempe, AZ 85287, USA
[7] School of Biological, Earth and Environmental Sciences, UNSW Sydney, Sydney, NSW, 2052, Australia
[8]UC Davis Institute for the Environment, UC Davis Earth and Planetary Sciences, University of California Davis, One Shields Avenue, Davis, CA 95616, USA
[9]Barcelona, Spain
[10]Federal Office for Radiation Protection, Koepenicker Allee 120, 10318 Berlin, Germany
[11]The Fredy & Nadine Herrmann Institute of Earth Sciences, The Hebrew University, The Edmond J. Safra Campus - Givat Ram, Jerusalem 9190401, Israel
[12]Department of Geography and Environmental Science, University of Reading, Reading RG6 6AH, UK
[13] Institute for Geological and Geochemical Research, HUN-REN Research, Research Centre for Astronomy and Earth Sciences, Budaörsi út 45, H-1112 Budapest, Hungary
[14]CSFK, MTA Centre of Excellence, Budapest, Konkoly Thege Miklós út 15-17, 1121 Budapest, Hungary
[15] Potsdam Institute for Climate Impact Research (PIK), Potsdam, Germany
[16] Alfred Wegener Institute (AWI), Helmholtz Centre for Polar and Marine Research, Potsdam, Germany.
[17]Department of Earth Sciences, ETH Zurich, Sonneggstrasse 5, 8092 Zurich, Switzerland
[18] Department of Geology, Institute of Geography and Earth Sciences, ELTE Eötvös Loránd University, Budapest, 1117, Hungary
[19]ANSTO, New Illawarra Road, Lucas Heights, NSW 2234, Australia
[20]Department of Geology, Trinity College Dublin, Dublin 2, Ireland
[21]Institute of Global Environmental Change, Xi'an Jiaotong University, Xi'an, Shaanxi, China 710049
[22]Institute of Geology, Innsbruck University, Innrain 52, Innsbruck 6020, Austria
[23]School of Environmental and Life Sciences, The University of Newcastle, NSW 2308, Australia

[24]Department of Geology, Faculty of Science, Chulalongkorn University, Bangkok 10330 Thailand

[25]University of Pisa, Department of Earth Sciences. Via Santa Maria 53, 56126 Pisa Italy

[26]College of Ocean and Earth Sciences, Xiamen University, Xiamen, Fujian, 361102, China

[27]School of Earth, Atmospheric and Life Sciences, University of Wollongong, Northfields Ave Wollongong, NSW 2522, Australia

[28]Department of Geological Engineering, Akdeniz University, 07100, Antalya, Türkiye

[29]Quaternary Geology, Department of Environmental Sciences, University of Basel, 4056, Switzerland

[30]Department of Climate Geochemistry, Max Planck Institute for Chemistry, Mainz, Germany

[31]Department of Geography and Environmental Sciences, Northumbria University, Newcastle upon Tyne, NE1 8ST, United Kingdom

[32]Department of Environmental Chemistry, Institute of Environmental Assessment and Water Research (IDAEA-CSIC), Jordi Girona, 18; 08034 Barcelona, Spain

[33]Inter-University Accelerator Centre, New Delhi 110067, India

[34]UMR IDEES 6266, CNRS, University of Rouen Normandy, 1, Rue Thomas Becket, Mont Saint-Aignan, 76130, France

[35]Center for Climate Physics, Institute for Basic Science, Busan, 46241, Republic of Korea

[36]Pusan National University, Busan, 46241, Republic of Korea

[37]Department of Earth, Environmental and Planetary Science, Brown University, Providence 02908, Rhode Island, USA

[38]Institute at Brown for Environment and Society, Brown University, Providence 02908, Rhode Island, USA

[39]Department of Earth, Atmospheric and Planetary Sciences, Massachusetts Institute of Technology, Cambridge, MA, USA

[40]Department of Geology and Geophysics, Woods Hole Oceanographic Institution, Woods Hole, MA, USA

[41]Ruprecht Karls University Heidelberg, Institute of Earth Sciences, Im Neuenheimer Feld 234, 69120 Heidelberg

[42]Ruprecht Karls University Heidelberg, Institute of Environmental Physics, Im Neuenheimer Feld 229, 69120 Heidelberg

[43]Department of Earth System Science, University of California Irvine, Croul Hall, Irvine, CA 92697-3100, USA

[+] A full list of authors appears at the end of the paper.

*These authors contributed equally to the manuscript.

*Correspondence to*: Nikita Kaushal (nikitageologist@gmail.com), Franziska Lechleitner (Franziska.lechleitner@unibe.ch), Micah Wilhelm (micah.wilhelm@wsl.ch)

**Abstract.** Paleoclimate information on multiple climate variables at different spatiotemporal scales is increasingly important to understand environmental and societal responses to climate change. A lack of high-quality reconstructions of past

hydroclimate has recently been identified as a critical research gap. Speleothems, with their precise chronologies, widespread distribution, and ability to record changes in local to regional hydroclimate variability, are an ideal source of such information. Here we present a new version of the Speleothem Isotopes Synthesis and AnaLysis database (SISALv3), which has been expanded to include trace element ratios and Sr-isotopes as additional, hydroclimate-sensitive geochemical proxies. The oxygen and carbon isotope data included in previous versions of the database have been substantially expanded. SISALv3,

contains speleothem data from 365 sites from across the globe, including 95 Mg/Ca, 85 Sr/Ca, 52 Ba/Ca, 25 U/Ca, 29 P/Ca and 14 Sr-isotope records. The database also has increased spatiotemporal coverage for stable oxygen (892) and carbon (620) isotope records compared to SISALv2 (673 and 430 stable oxygen and carbon records, respectively). Additional meta information has been added to improve machine-readability and filtering of data. Standardized chronologies are included for all new entities together with the originally published chronologies. The SISALv3 database thus constitutes a unique resource

of speleothem paleoclimate information that allows regional-to-global paleoclimate analyses based on multiple geochemical

proxies, allowing more robust interpretations of past hydroclimate and comparisons with isotope-enabled climate models and other earth system and hydrological models. The database can be accessed at http://dx.doi.org/10.5287/ora-2nanwp4rk.

## 1 Introduction

Speleothems, secondary cave carbonate precipitates, are a rich paleoenvironmental archive of geochemical data (Wong and Breecker, 2015). Due to their widespread distribution (Comas-Bru et al., 2020) and their precise chronologies (Henderson, 2006), they can provide paleoclimate data at seasonal (Baldini et al., 2021) to multi-annual resolution spanning millennial and longer time scales (Cheng et al., 2016; Stoll et al., 2022).

The Speleothem Isotopes Synthesis and AnaLysis working group (SISAL WG) is an international effort to synthesize speleothem data under the umbrella of the Past Global Changes (PAGES) project (Comas-Bru et al., 2017; Comas-Bru and Harrison, 2019). The SISAL WG aims to answer critical open questions in paleoclimate science with a focus on regional to global trends and event synchronization. To address these questions, the SISAL WG has been developing standardized and quality checked databases. The first three versions of the database (SISALv1, SISALv1b and SISALv2) provided the paleoclimate community with a growing resource of speleothem geochemical data (Atsawawaranunt et al., 2018; Comas-Bru et al., 2020, 2019), specifically oxygen ($\delta^{18}O$) and carbon ($\delta^{13}C$) isotope records, and age-model ensembles, along with an online tool -the SISAL webApp - to increase accessibility to the SISAL database (Hatvani et al., 2024). The SISAL database versions have been exploited (i) to better understand the drivers of speleothem environmental proxies and improve their interpretations (Baker et al., 2019, 2021; Fohlmeister et al., 2020; Treble et al., 2022; Skiba and Fohlmeister, 2023), (ii) to provide a resource on the interpretation of speleothem records at a regional level, identifying key gaps and future work (Kaushal et al., 2018; Lechleitner et al., 2018; Braun et al., 2019a; Burstyn et al., 2019; Deininger et al., 2019; Kern et al., 2019; Oster et al., 2019; Zhang et al., 2019; Lorrey et al., 2020), and (iii) to understand the mechanisms of past climate change including through comparison with isotope-enabled climate models (Comas-Bru et al., 2019; Parker et al., 2021a; Bühler et al., 2022; Parker and Harrison, 2022; Parker et al., 2021b) and other modelling approaches (Skiba et al., 2023).

The new SISALv3 database provides an increased dataset of oxygen and carbon isotope data, interpreted as records of hydroclimate and vegetation dynamics/bioproductivity (Wong and Breecker, 2015), and has been significantly expanded to include data on Sr, Mg, Ba, U, typically tracers for hydrological processes in the karst and cave (Fairchild et al., 2000; Johnson et al., 2006; Fairchild and Treble, 2009; Wassenburg et al., 2016), and P, recognized as tracer for surface bioproductivity (Treble et al., 2003; Borsato et al., 2007; McDonough et al., 2022) (Table 1). Also included are data on Sr-isotopes, as these are an important proxy for hydroclimatic processes and may provide information on local hydrology and soil source, production and/or erosion (e.g., Li et al., 2005; Ünal-İmer et al., 2016; Wortham et al., 2017; Weber et al., 2018; Ward et al., 2019; Utida et al., 2020). Ratios of Sr/Ca, Mg/Ca, Ba/Ca and U/Ca, coupled with $\delta^{13}C$ information, are sensitive to water-rock interactions and residence time (Fairchild et al., 2000; Johnson et al., 2006). An important mechanism that drives variability in these multiple proxies in quantifiable ways is the process of prior calcite/carbonate precipitation (PCP), through which

carbonate precipitated along flow paths in the karst and on the cave roof will lead to altered element concentration in cave drip waters from which the speleothem ultimately precipitates (Fairchild et al., 2000; Day and Henderson, 2013). An increase in

PCP usually occurs in times of drought that facilitate increased water-rock residence times and degassing in the karst (Fairchild et al, 2000). The strength of these proxies is that they provide robust climatic and environmental information via a multi-proxy approach that will need to be tailored for different karst and climatic settings (Table 1). The SISAL Working Group is currently working on projects with the new additional proxies to explore and gain more detailed insights. We provide examples of proxy interpretations with linked references, but we must emphasize that this list is not exhaustive, the interpretations are time-scale

dependent, and in most cases, multi-proxy approaches are necessary (Table 1). The SISALv3 database, augmented with trace element proxies thus provides a multi-proxy dataset that can be used for long-term drought reconstructions in the past, and to better understand the forcings, mechanisms and periodicities of such events. In addition to the new geochemical data, extensive metadata including information on parameters such as vegetation and karst type as well as entity (i.e. speleothem dataset) images are provided to aid robust interpretations.

The SISALv3 database will allow the systematic and global analysis of stable isotope and trace element variability, and elucidate how trace element data can be used to strengthen climatic interpretations from speleothem oxygen ($\delta^{18}O$) and carbon ($\delta^{13}C$) records. The database can be accessed at http://dx.doi.org/10.5287/ora-2nanwp4rk.

| Proxy | Potential drivers | Selected relevant references |
|---|---|---|
| $\delta^{18}O$ | Semi-quantitative temperature reconstruction dependent on the combined effect of temperature dependency of meteoric precipitation $\delta^{18}O$ and in-cave temperature on carbonate $\delta^{18}O$ | (Dorale et al., 1998; Mangini et al., 2005; Moseley et al., 2015; Koltai et al., 2017; Wendt et al., 2021; Luetscher et al., 2021; Wolf et al., 2024; Wainer et al., 2011) |
| | Change in source water composition e.g. as tracers of ice sheet meltwaters during deglaciations | (Stoll et al., 2022; Badertscher et al., 2011; Frumkin et al., 1999; Meckler et al., 2012) |
| | Change in moisture transport trajectory or change in moisture source (sometimes linked to seasonality) | (Lachniet et al., 2014; Cheng et al., 2016; Luetscher et al., 2021; Frumkin et al., 1999) |
| | Change in seasonality of precipitation e.g. increase winter rain versus summer rain in different climate states | (Cheng et al., 2009b; Baldini et al., 2019; Cheng et al., 2019; Wang et al., 2001) |
| | Precipitation amount at the cave site and upstream rainout | (Bar-Matthews et al., 2003; Hu et al., 2008; Cheng et al., 2016; Columbu et al., 2019; Cheng et al., 2013) |
| | Large-scale circulation and supra-regional climate e.g. the Indian Summer Monsoon | (Cheng et al., 2016; Kathayat et al., 2016) |

| | | |
|---|---|---|
| | Recharge processes and karst flow paths | (Ayalon et al., 1998; Baker et al., 2019; Treble et al., 2022) |
| $\delta^{13}C$ | Semi-quantitative temperature reconstruction linked to changes in vegetation and soil respiration (requires additional proxies e.g. Mg/Ca) | (Genty et al., 2003, 2006; Lechleitner et al., 2021; Stoll et al., 2022, 2023) |
| | Vegetation density variability e.g. low vegetation density zones versus high vegetation density zones | (Fohlmeister et al., 2020) |
| | Metabolic pathway e.g. C3 versus C4 pathway | (Baker et al., 1997) |
| | Hydroclimate through the prior calcite precipitation mechanism and/or drip rate changes (requires additional proxies e.g. Mg/Ca) | (Johnson et al., 2006; Owen et al., 2016; Carolin et al., 2019b; Fairchild et al., 2000) |
| Mg/Ca | Hydroclimate through the prior calcite precipitation mechanism and/or drip rate changes, potential for semi-quantitative precipitation reconstruction (requires additional proxies e.g. $\delta^{13}C$, Sr/Ca and Ba/Ca, and/or cave monitoring data) | (Johnson et al., 2006; Owen et al., 2016; Carolin et al., 2019b; Fairchild et al., 2000; Warken et al., 2018) |
| | Hydroclimate through dust activity and marine aerosol input (requires additional proxies e.g. Na/Ca, Sr/Ca and Ba/Ca, and/or cave monitoring data) | (Faraji et al., 2023; Carolin et al., 2019b) |
| | Hydroclimate through water residence time in soil and karst | (Roberts et al., 1998; Treble et al., 2003; Tremaine and Froelich, 2013) |
| Sr/Ca | Hydroclimate through the prior calcite precipitation mechanism and/or drip rate changes (requires additional proxies e.g. $\delta^{13}C$, Mg/Ca and Ba/Ca, and/or cave monitoring data) | (Johnson et al., 2006; Owen et al., 2016; Carolin et al., 2019b; Fairchild et al., 2000) |
| | Aeolian transport (increased confidence in interpretation with $^{87}Sr/^{86}Sr$ data) | (Goede et al., 1998) |
| Ba/Ca | Hydroclimate through the prior calcite precipitation mechanism and/or drip rate changes (requires additional proxies e.g. $\delta^{13}C$, Mg/Ca and Sr/Ca, and/or cave monitoring data) | (Johnson et al., 2006) |
| | Growth rate | (Treble et al., 2003) |
| | Soil mineral weathering | (Riechelmann et al., 2020; Rutlidge et al., 2014) |

| U/Ca | Hydroclimate through the prior aragonite precipitation mechanism and/or drip rate changes (requires additional proxies e.g. $\delta^{13}C$, Mg/Ca, Ba/Ca and Sr/Ca, and/or cave monitoring data) | (Jamieson et al., 2016) |
| --- | --- | --- |
| | Enhanced infiltration via complexes (e.g., uranyl-phosphate) (requires additional proxies e.g. P/Ca) | (Treble et al., 2003) |
| P/Ca | Biomass cycling including wildfire; enhanced infiltration via complexes | (Huang et al., 2001; Treble et al., 2003; Borsato et al., 2007; McDonough et al., 2022) |
| Sr isotopes | Hydroclimate through proportional source changes | (Verheyden et al., 2000; Utida et al., 2020) |
| | Aeolian activity | (Li et al., 2005) |

*Table 1: Summary of speleothem geochemical proxies included in SISALv3, examples for their possible interpretations, and relevant references.*

## 2 Data and Methods

### 2.1 New data formatting and processing

All trace elements are reported normalized as ratios with respect to Ca (X/Ca, where X stands for the individual elements) in units of mmol/mol. In the following manuscript, "trace element" refers to the normalized ratio to Ca. A standardized conversion sheet is used to facilitate conversions from gram to mol units (available in the repository). Sr-isotope data is reported as $^{87}Sr/^{86}Sr$ values. For internal consistency, and to facilitate future intercomparison and synthesis studies, the measurement method and reference materials used, and measurement precision are also reported for both trace elements and Sr-isotopes. Mechanisms relevant to hydroclimate interpretations from speleothems are based on a multi-proxy approach of stable isotopes and one or more trace element ratios. Therefore, the SISALv3 database structure allows for trace element measurements to be added at the depths of the stable isotope measurements on a given entity. However, between 35 and 86% of the records (depending on element) were measured using in-situ techniques, such as laser ablation inductively coupled plasma mass spectrometry (LA-ICP-MS), and these datasets are typically generated at a higher resolution (10-100 μm) than the stable isotope records (Jochum et al., 2012). These data have been downsampled to the resolution of the stable isotope data for the same speleothem. Downsampling was performed by computing averages (and standard deviations) of the trace element measurements for corresponding stable isotope sampling depths. This implicitly assumes that the same or a (depth-equivalent) parallel sampling track were used for trace elements and stable isotopes and that the isotope sampling was continuous. Downsampling allows the trace element data to be represented by the same depth-age model as the stable isotope record. For records submitted by the authors where the originally published dataset was at higher resolution than reported in the SISAL

database, standardized *.txt datafiles are also available in the repository (see section 5.3 on Code and Data accessibility). No new chronological information or separate age models are reported for these datasets.

## 2.2 Additional metadata

New metadata fields are included in the Entity table (see database structure; Figure 1) to allow users to select sites with similar environmental conditions and to take account of factors that might influence the interpretation of individual records. These include information on vegetation, land use, land cover, and host rock type above the cave. This information is often missing from publications and was not available from data contributors, so information from data products has been added as additional fields to the database for completeness. Vegetation type and land use information were provided by the original investigators. Additionally, information on land use and land cover was taken from the Copernicus Global Land Service Land Cover database (LCC v3.0.1 Epoch 2019; (Buchhorn et al., 2021, 2020), extracted with a radius of 250 m around the cave site. Information on the carbonate/evaporite host rock at the cave sites was taken from the WOKAM database (Goldscheider et al., 2020) extracted with a radius of 1000 m.

The database also indicates if the trace element content of the host rock and drip water feeding the speleothem is available (but does not include the actual values). Drip height (i.e., the distance the drip falls from the ceiling of the cave to the speleothem), and the difference between dripwater and carbonate $\delta^{18}O$ values are given, based on information provided by the original investigators.

The SISAL WG repository now hosts images of the entities (speleothem sections) and maps of cave sites. These allow users to evaluate petrographic features that may influence the trace element and stable isotopic records and to check whether cave morphology could potentially influence the climate in the cave (Covington and Perne, 2015). The entity table in the database contains fields indicating whether maps and images are available.

## 2.3 Changes to database structure

The structure of the SISALv3 database (Figure 1) has been changed to accommodate additional data and metadata, and to optimize the organization of information, as described below.

### 2.3.1 New geochemical data and metadata fields

The elemental ratio for each trace element and the Sr-isotope data are given in individual tables that contain sample identifiers (*sample_id*), the measurement value and the measurement precision. The *sample_id* provides the link to the Sample table and thus links these data to the stable isotope data (Figure 1, Table 2).

Metadata for the measurements are stored in the Entity table. For each elemental ratio (Sr/Ca, Mg/Ca, U/Ca, Ba/Ca and P/Ca), the Entity table indicates whether the data are available ("yes/no/other/unknown"), the measurement method, the laboratory reference materials used and, where applicable, the downsampling methods used. The table also indicates if high-resolution

trace element data is available. For Sr-isotopes, the Entity table specifies whether this dataset is available, what measurement method was employed and how the measurement was standardized (Figure 1, Table 2).

SISALv3 now provides a unique, persistent identifier for each speleothem (*persist_id*) in the Entity table (Figure 1, Table 2). This was needed because there was an increasing issue with non-unique entity names, and to deal with the fact that different datasets from the same stalagmite had different *entity_id*s (e.g., for datasets covering different time periods in the same speleothem). Thus, the field *entity_id* provides a unique identifier for a specific dataset, but not necessarily for a specific speleothem, while the *persist_id* uniquely identifies the speleothem. The *persist_id* were created by combining the *site_id* and *entity_name* (without special characters). There are 838 unique *persist_id*s and 902 unique *entity_id*s in the database.

### 2.3.2 Changes in existing database fields and options

The fields "geology" and "rock age" were moved from the Site table to the Entity table (Figure 1, Table 2). This was done to allow for variability in these parameters within the same cave system, particularly relevant for the interpretation of $\delta^{13}$C and trace element data. The field "trace elements" (yes/no) in the Entity table was removed as it is now redundant. The field "iso_std" describing the reference material used for measurement of $\delta^{18}$O and $\delta^{13}$C values was moved from the stable isotope tables to the Entity metadata table. A number of options for entries in the metadata fields were changed (Table 3). The majority of these changes were additions to the previously available options in light of the entries made in the 'Notes' section to allow for more "metadata-filterable" database mining. A few options were removed from the metadata fields since they have never been used in previous database versions.

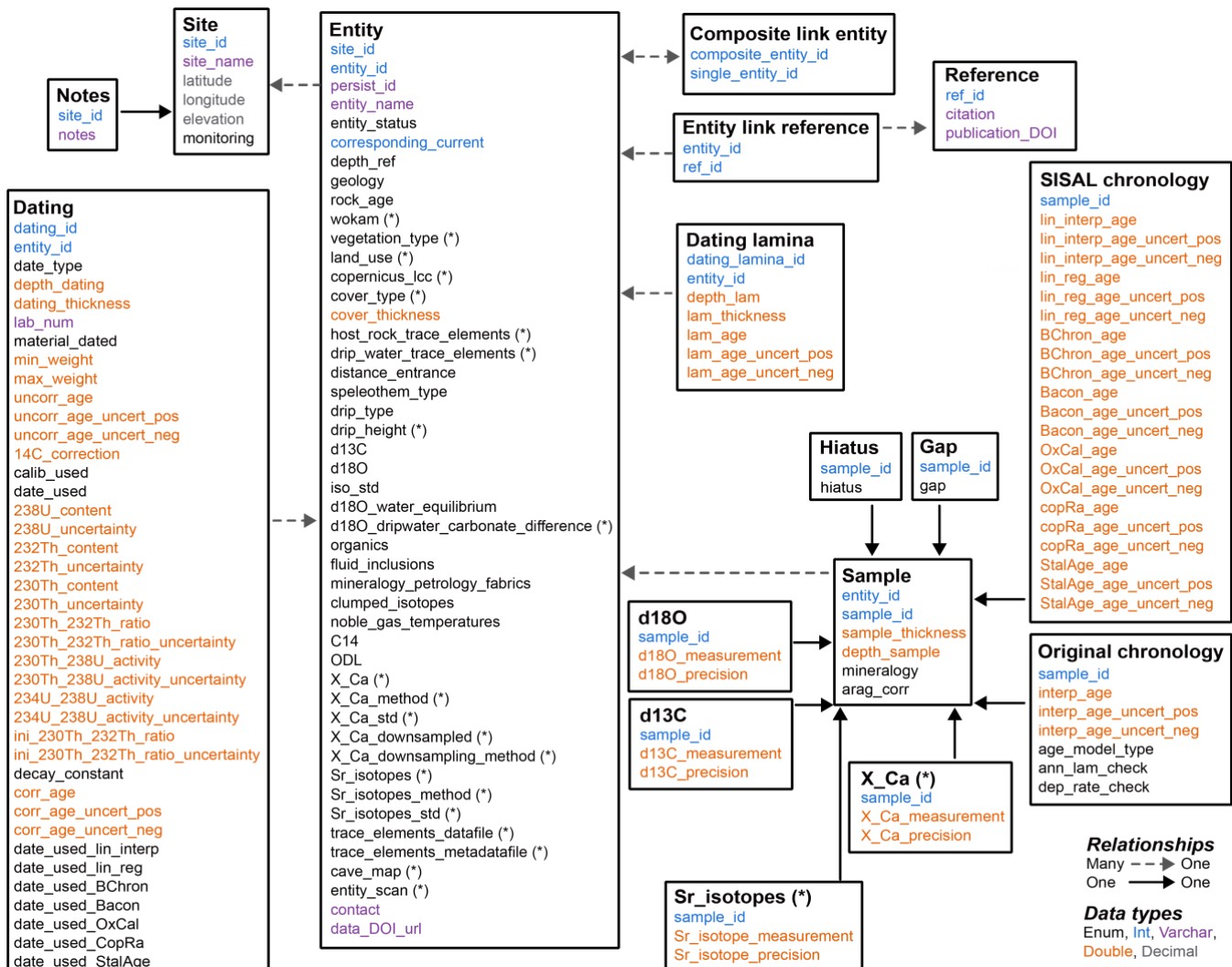

*Figure 1: Structure of the SISALv3 database. Fields and tables marked with (*) refer to new information added in SISALv3; see table 2 for details. The colors refer to the format of that field: Enum, Int, Varchar, Double or Decimal. More information on the list of pre-defined menus can be found in the Supplementary information (Table S1). For trace element records, a series of identical tables was generated (labelled X_Ca where X stands for the specific element: Mg, Sr, Ba, U, P).*

| Action | Field label | Description | Format | Constraints |
|---|---|---|---|---|
| **Changes made to the SITE table** | | | | |
| Field removed | *geology* | | | |
| Field removed | *rock_age* | | | |
| | | | | |

| **Changes made to the ENTITY table** | | | | |
|---|---|---|---|---|
| Field added | *persist_id* | persistent, unique identifier for each speleothem | Text | |
| Field added | *geology* | Information on geology | Text | Selection from predefined list |
| Field added | *rock_age* | Information on bedrock age | Text | Selection from predefined list |
| Field added | *wokam* | Information on type of carbonate/evaporite rock from WOKAM database | Text | Added by SISAL SC at database level |
| Field added | *vegetation_type* | Information on vegetation cover | Text | Selection from predefined list |
| Field added | *land_use* | Information on land use (publication/data contributors) | Text | Selection from predefined list |
| Field added | *copernicus_lcc* | Information on land cover from Copernicus land cover classification dataset | Text | Added by SISAL SC at database level |
| Field added | *cover_type* | Information on land cover (publication/data contributors) | Text | Selection from predefined list |
| Field added | *host_rock_trace_elements* | Indication whether trace element data from the host rock has been measured | Text | Selection from predefined list |
| Field added | *drip_water_trace_elements* | Indication whether trace element data from the drip water has been measured | Text | Selection from predefined list |
| Field added | *drip_height* | Information on drip height (in m) | Numeric | Free to fill |
| Field added | *iso_std* | Information on reference material used for oxygen and carbon isotope measurements | Text | Selection from predefined list |
| Field added | *d18O_dripwater_carbonate_difference* | Information on difference between dripwater and carbonate oxygen isotope values | Numeric | Free to fill |
| Field removed | *trace_elements* | | | |
| Field added | *Sr_Ca* | Indication whether Sr/Ca data has been measured | Text | Selection from predefined list |
| Field added | *Sr_Ca_method* | Information on measurement method for Sr/Ca | Text | Selection from predefined list |

| Field added | Sr_Ca_std | Information on reference material used for Sr/Ca measurements | Text | Selection from predefined list |
|---|---|---|---|---|
| Field added | Sr_Ca_downsampled | Information on whether Sr/Ca data had to be downsampled | Text | Selection from predefined list |
| Field added | Sr_Ca_downsampling_method | Information on downsampling method for Sr/Ca, if applicable | Text | Selection from predefined list |
| Field added | Mg_Ca_method | Information on measurement method for Mg/Ca | Text | Selection from predefined list |
| Field added | Mg_Ca_std | Information on reference material used for Mg/Ca measurements | Text | Selection from predefined list |
| Field added | Mg_Ca_downsampled | Information on whether Mg/Ca data had to be downsampled | Text | Selection from predefined list |
| Field added | Mg_Ca_downsampling_method | Information on downsampling method for Mg/Ca, if applicable | Text | Selection from predefined list |
| Field added | Ba_Ca | Indication whether Ba/Ca data has been measured | Text | Selection from predefined list |
| Field added | Ba_Ca_method | Information on measurement method for Ba/Ca | Text | Selection from predefined list |
| Field added | Ba_Ca_std | Information on reference material used for Ba/Ca measurements | Text | Selection from predefined list |
| Field added | Ba_Ca_downsampled | Information on whether Ba/Ca data had to be downsampled | Text | Selection from predefined list |
| Field added | Ba_Ca_downsampling_method | Information on downsampling method for Ba/Ca, if applicable | Text | Selection from predefined list |
| Field added | U_Ca | Indication whether U/Ca data has been measured | Text | Selection from predefined list |
| Field added | U_Ca_method | Information on measurement method for U/Ca | Text | Selection from predefined list |
| Field added | U_Ca_std | Information on reference used for U/Ca measurements | Text | Selection from predefined list |
| Field added | U_Ca_downsampled | Information on whether U/Ca data had to be downsampled | Text | Selection from predefined list |

| | | | | |
|---|---|---|---|---|
| Field added | *U_Ca_downsampling_method* | Information on downsampling method for U/Ca, if applicable | Text | Selection from predefined list |
| Field added | *P_Ca* | Indication whether P/Ca data has been measured | Text | Selection from predefined list |
| Field added | *P_Ca_method* | Information on measurement method for P/Ca | Text | Selection from predefined list |
| Field added | *P_Ca_std* | Information on reference material used for P/Ca measurements | Text | Selection from predefined list |
| Field added | *P_Ca_downsampled* | Information on whether P/Ca data had to be downsampled | Text | Selection from predefined list |
| Field added | *P_Ca_downsampling_method* | Information on downsampling method for P/Ca, if applicable | Text | Selection from predefined list |
| Field added | *Sr_isotopes* | Indication whether Sr isotope data has been measured | Text | Selection from predefined list |
| Field added | *Sr_isotopes_method* | Information on measurement method for Sr isotopes | Text | Selection from predefined list |
| Field added | *Sr_isotopes_std* | Information on reference material used for Sr isotope measurements | Text | Selection from predefined list |
| Field added | *trace_elements_datafile* | Information on whether the original trace elements data is available in the repository | Text | Selection from predefined list |
| Field added | *trace_elements_metadatafile* | Information whether original trace element metadata is available in the repository | Text | Selection from predefined list |
| Field added | *cave_map* | Information whether a copy of cave map is available in the repository | Text | Selection from predefined list |
| Field added | *entity_scan* | Information whether a scan of the speleothem in the repository | Text | Selection from predefined list |
| | | | | |
| **Changes made to the d18O and d13C tables** | | | | |

| | | | | | |
|---|---|---|---|---|---|
| Field removed | *iso_std* | Information on reference material used for $\delta^{18}O$ and $\delta^{13}C$ measurements | Text | Selection from predefined list | |

*Table 2: Changes made to the Site, Entity and stable isotope tables compared to SISALv2.*

| Table name | Action | Field label | Reason | Format | Constraints |
|---|---|---|---|---|---|
| Entity | Added "mixed (see notes)" option | *speleothem_type* | Standardisation of option across fields | Text | Selected from pre-defined list |
| Entity | Added "other (see notes)" option | *speleothem_type* | Standardisation of option across fields | Text | Selected from pre-defined list |
| Entity | Removed "magmatic" option | *geology* | Not used | Text | Selected from pre-defined list |
| Entity | Removed "granite" option | *geology* | Not used | Text | Selected from pre-defined list |
| Entity | Added "dolomite limestone" option | *geology* | Machine-readable format option | Text | Selected from pre-defined list |
| Entity | Added "marly limestone" option | *geology* | Machine-readable format option | Text | Selected from pre-defined list |
| Entity | Added "calcarenite" option | *geology* | Machine-readable format option | Text | Selected from pre-defined list |
| Entity | Added "other (see notes)" option | *geology* | Option to include free text in notes | Text | Selected from pre-defined list |
| Entity | Added "other (see notes)" option | *rock_age* | Option to include free text in notes | Text | Selected from pre-defined list |
| Entity | Added "mixed (see notes)" option | *rock_age* | Reflect overburden with rocks of different ages | Text | Selected from pre-defined list |
| Entity | Removed "mixture" | *drip_type* | Replaced with "mixed (see notes)" option | Text | Selected from pre-defined list |
| Entity | Removed "not applicable" | *drip_type* | Not used | Text | Selected from pre-defined list |
| Entity | Added "mixed (see notes)" option | *drip_type* | Standardisation of option across fields | Text | Selected from pre-defined list |

| Entity | Added "other (see notes)" option | *drip_type* | Option to include free text in notes | Text | Selected from pre-defined list |
|---|---|---|---|---|---|
| Entity | Added "other (see notes)" option | *d18O_water_equilibrium* | Option to include free text in notes | Text | Selected from pre-defined list |
| Entity | Added "other (see notes)" option | *organics* | Option to include free text in notes | Text | Selected from pre-defined list |
| Entity | Added "other (see notes)" option | *fluid_inclusions* | Option to include free text in notes | Text | Selected from pre-defined list |
| Entity | Added "other (see notes)" option | *mineralogy_petrology_fabric* | Option to include free text in notes | Text | Selected from pre-defined list |
| Entity | Added "other (see notes)" option | *clumped_isotopes* | Option to include free text in notes | Text | Selected from pre-defined list |
| Entity | Added "other (see notes)" option | *noble_gas_temperatures* | Option to include free text in notes | Text | Selected from pre-defined list |
| Entity | Added "other (see notes)" option | *C14* | Option to include free text in notes | Text | Selected from pre-defined list |
| Entity | Added "other (see notes)" option | *ODL* | Option to include free text in notes | Text | Selected from pre-defined list |
| Dating | Added "mixed (see notes)" option | *material_dated* | Align with options in the sample table | Text | Selected from pre-defined list |
| Dating_lamina | Added "Year of chemistry" option | *modern_reference* | Align with options in the dating table | Text | Selected from pre-defined list |
| Sample | Removed "secondary calcite" option | *mineralogy* | Not used | Text | Selected from pre-defined list |
| Sample | Removed "vaterite" option | *mineralogy* | Not used | Text | Selected from pre-defined list |
| Sample | Added "organic" option | *mineralogy* | Addition of option | Text | Selected from pre-defined list |
| Sample | Added "other (see notes)" option | *mineralogy* | Option to include free text in notes | Text | Selected from pre-defined list |
| Sample | Removed "combination of methods" option | *age_model_type* | Standardisation of option across fields | Text | Selected from pre-defined list |

| Sample | Added "mixed (see notes)" option | *age_model_type* | Standardisation of option across fields | Text | Selected from pre-defined list |
|---|---|---|---|---|---|
| Sample | Added "other (see notes)" option | *dep_rate_check* | Option to include free text in notes | Text | Selected from pre-defined list |

*Table 3: Changes made to the predefined options for metadata fields compared to SISALv2.*

## 3 Quality control

The SISAL WG has used several levels of quality control (QC) and this practice was continued for SISALv3 (Figure 2). The initial data compilation is performed by SISAL regional coordinators and/or in liaison with the data contributors into
215 standardized excel workbooks. The first QC level consists in the expert assessment by the SISAL regional coordinators who double-check completeness of entered data and correctness of measurement units where applicable. Standardized unit conversion sheets for common conversions (e.g. degrees-minutes-seconds to decimal degrees for site information; atomic ratios to activity ratios for dating information; mg/g to mmol/mol for trace element-to-Ca ratios for trace element information) have been provided to regional coordinators (see repository). The completed workbook(s) are subjected to a series of automated
QC (e.g., age model matches discreet dating information, hiatuses are placed at the correct depth) by the database managers. When the datasets pass automated QC, and no further corrections are necessary, the dataset workbook and auto generated QC figures are sent to the data contributors for final evaluation and approval. The same workflow has been followed for the *.txt trace element datafiles. The new metadata fields of vegetation_type and land_use that have been added to SISALv3 for entities that were included in SISALv2 from publications. Data already included in SISALv2 has been checked, and mistakes /
unknowns identified during previous data analysis or during the process of trace element data addition were corrected. A comprehensive summary of the changes made to existing entities between SISALv2 and SISALv3 is shown in Table 4.

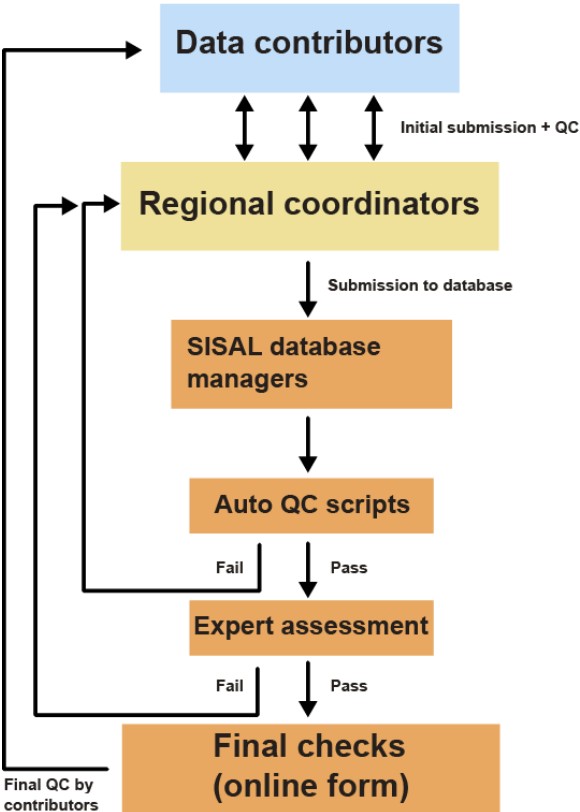

*Figure 2: Quality checking workflow adopted for inclusion of datasets in SISAL. The colors indicate different quality check levels: blue - data contributing sources (original authors or datasets deposited in repositories and publication supplementary information); yellow - SISAL regional coordinator group with regional expertise; orange - SISAL database managers.*

| Modification | v2 to v3 |
|---|---|
|  |  |
| **Site table** |  |
| Number of new sites | 72 |
| Pre-existing sites with new entities | 37 |
|  |  |
| **Entity table** |  |
| Number of new entities | 211 |
| Entities added to pre-existing sites | 71 |
| Entities with updated entity.entity_status | 33 |
| Entities with altered entity.corresponding_current | 1 |
| Entities with altered geology | 106 |

| | |
|---|---|
| Entities with altered rock_age | 58 |
| Entities with altered entity.cover_thickness | 6 |
| Entities with altered distance_entrance | 1 |
| Entities with altered d13C | 109 |
| Entities with altered d18O | 15 |
| Entities with altered organics | 9 |
| Entities with altered fluid_inclusions | 9 |
| Entities with altered mineralogy_petrology_fabric | 14 |
| Entities with altered clumped_isotopes | 9 |
| Entities with altered noble_gas_temperatures | 12 |
| Entities with altered C14 | 3 |
| Entities with altered ODL | 7 |
| Entities with altered contact | 62 |
| Entities with altered data_DOI_url | 20 |
| | |
| **Dating table** | |
| Addition of "event: hiatus" to an entity | 1 |
| Changes in hiatus depths | 1 |
| Changes in depths of "Event: start/end of laminations" | 1 |
| Alterations in dating.date_type | 2 |
| Alterations in dating.depth_dating | 5 |
| Alterations in dating.material_dated | 2 |
| Alterations in dating.min_weight | 6 |
| Alterations in dating.max_weight | 6 |
| Alterations in dating.uncorr_age | 15 |
| Alterations in dating.uncorr_age_uncert_pos | 13 |
| Alterations in dating.uncorr_age_uncert_neg | 13 |
| Alterations in dating.date_used | 27 |
| Alterations in dating.238U_content | 89 |
| Alterations in dating.238U_uncertainty | 34 |
| Alterations in dating.232Th_content | 96 |
| Alterations in dating.232Th_uncertainty | 85 |
| Alterations in dating.230Th_232Th_ratio | 206 |
| Alterations in dating.230Th_232Th_ratio_uncertainty | 200 |
| Alterations in dating.230Th_238U_activity | 24 |

| | |
|---|---|
| Alterations in dating.230Th_238U_activity_uncertainty | 21 |
| Alterations in dating.234U_238U_activity | 381 |
| Alterations in dating.234U_238U_activity_uncertainty | 433 |
| Alterations in dating.ini_230Th_232Th_ratio | 519 |
| Alterations in dating.ini_230Th_232Th_ratio_uncertainty | 485 |
| Alterations in dating.decay_constant | 77 |
| Alterations in dating.corr_age | 71 |
| Alterations in dating.corr_age_uncert_pos | 26 |
| Alterations in dating.corr_age_uncert_neg | 28 |
| | |
| **Sample table** | |
| Altered sample.depth_sample | 1084 |
| Altered sample.mineralogy | 294 |
| Altered sample.arag_corr | 294 |
| Entities that had d18O time series altered (changes in depth/ duplicate isotope values) | 4 |
| Entities that had d13C time series altered (changes in depth/ duplicate isotope values) | 2 |
| | |
| **Original chronology** | |
| Altered original_chronology.interp_age | 7440 |
| | |
| **References** | |
| How many entities had changes in references? | 100 |
| How many citations have a different pub_DOI? | 100 |
| | |
| **Notes** | |
| Sites with notes modified | 121 |

*Table 4: Summary of the modifications applied to records in version 2 (Comas-Bru et al., 2020) of the SISAL database. Note that the changes in the dating and sample table were counted by dating_id and sample_id, respectively, which leads to a large number of changes.*

## 4 Overview of database contents

### 4.1 Trace element and Sr-isotope records

SISALv3 contains 95 Mg/Ca, 85 Sr/Ca, 52 Ba/Ca, 25 U/Ca, 29 P/Ca and 14 Sr-isotope records (Table 5). This corresponds to
~60% of the known published data, based on an assessment by the SISAL WG. There is a clear regional bias in the database
with European entities dominating every elemental ratio (Figure 3). The Sr-isotope records are more evenly distributed, with
records from every region except Asia and Oceania. Temporal coverage for the combined trace element and Sr-isotope dataset
is high during the last 2000 years (~60 entities per 20 year interval) and the Holocene (~60 entities per 250 year interval), and
then drops to 20-40 entities per 10,000 year interval for the last glacial cycle (12-120 ka BP, where ka stands for 1000 years
and BP for "before present", defined as 1950, Figure 4). Beyond ~120 ka BP, the number of entities gradually decreases until
the U-Th dating limit is reached (~640 ka).

Where the original measured laser ablation data have been provided by data contributors, these have been made available as
*.txt datafiles in the repository (Table 5). Forty-six trace element records (Mg/Ca: 15, Sr/Ca: 17, Ba/Ca: 4, U/Ca: 5, P/Ca: 5,
Sr-isotopes: 2) are only provided in the original format (*.txt files), either because they could not be converted to mmol/mol
or because the trace element data were not measured at stable isotope equivalent depths and were at an insufficiently high
resolution for accurate resampling. Additional elements that are not included in the database, but have been submitted by data
contributors, are also provided as *.txt files (e.g., Mn, Fe, Zn, Th, Pb, K, Na).

| Geochemical data | Total number in SISAL | Downsampled by original authors | Downsampled by SISAL | Only in repository |
|---|---|---|---|---|
| Mg/Ca | 95 | 12 | 15 | 15 |
| Sr/Ca | 85 | 11 | 11 | 17 |
| Ba/Ca | 52 | 11 | 9 | 4 |
| U/Ca | 25 | 2 | 6 | 5 |
| P/Ca | 29 | 4 | 12 | 5 |

*Table 5: Summary of number of trace element records in SISAL and downsampling methods applied.*

**4.2 New stable isotope records**

SISALv3 provides a significantly expanded oxygen isotope dataset compared to SISALv2 (Table 6, Table 7, Figure 5), with 892 $\delta^{18}$O records from 365 sites, compared to 673 records in SISALv2. The most significant increases in $\delta^{18}$O records are in Africa (+28 records), Europe (+73), and the Middle-East (+17; Table 6). SISALv3 contains 334 entities covering the last 2000
260   years, of which 78 are new (Figure 6). As record density begins to decrease with age (Figure 6), the spatial distribution is reduced as well. For the Last Glacial Maximum (20 – 22 ka BP), SISALv3 contains 92 entities (11 new), while for the Last Interglacial (124 – 126 ka BP), 66 entities are available (15 new). Four $\delta^{18}$O records previously included in SISALv2 have been modified to correct previous mistakes (Table 4), these are *entity_id*s 110 (CUR4; Novello et al., 2016), 169 (Dim-E3; Ünal-İmer et al., 2015), 447 (JAR4; Novello et al., 2017), and 573 (Gej-1; Flohr et al., 2017).

There has also been a significant increase in the number of $\delta^{13}$C records added, with 620 records in SISALv3 compared to 430 in SISALv2 (Table 6, Figure 7). At the regional scale, the most significant increases in $\delta^{13}$C records is for Africa (+23 records), Asia (+33), and Europe (+66, Table 6). The $\delta^{13}$C record coverage decreases following the same patterns as the trace elements and $\delta^{18}$O records (Figure 8). Two $\delta^{13}$C records previously included in SISALv2 have been modified to correct previous mistakes (Table 4), these are *entity_id*s 169 (Dim-E3; Ünal-İmer et al., 2015), and 573 (Gej-1; Flohr et al., 2017).

| Region | $\delta^{18}$O records in v3 | Increase compared to v2 (counts) | $\delta^{13}$C records in v3 | Increase compared to v2 (counts) |
|---|---|---|---|---|
| Africa | 73 | 28 | 63 | 23 |
| Asia | 237 | 50 | 105 | 33 |
| Europe | 243 | 73 | 213 | 66 |
| Middle East | 60 | 17 | 43 | 14 |
| Oceania | 100 | 11 | 66 | 11 |
| North and Central America | 88 | 9 | 72 | 9 |
| South America | 97 | 21 | 54 | 13 |

*Table 6: Summary of the new $\delta^{18}$O and $\delta^{13}$C records added to SISALv3 compared to SISALv2.*

| site_id | site_name | region | latitude | longitude | persist_id | entity_id | entity_name | citation |
|---|---|---|---|---|---|---|---|---|
| 70 | Abaco Island cave | Bahamas | 26.23 | -77.16 | 70-ABDC12 | 692 | AB-DC-12_2023 | (Arienzo et al., 2017) |
| 79 | Dim cave | Turkey | 36.534 | 32.1056 | 79-DIME4 | 693 | Dim-E4_2023 | (Ünal-İmer et al., 2016, 2015) |

| | | | | | | | |
|---|---|---|---|---|---|---|---|
| | | | | | 79-DIM1 | 754 | Dim1 | (Rowe et al., 2020) |
| 144 | Botuverá cave | Brazil | -27.2247 | -49.1569 | 144-BT2 | 694 | BT-2_2007 | (Cruz et al., 2007) |
| 145 | Antro del Corchia | Italy | 43.9833 | 10.2167 | 145-CD31 | 695 | CD3-1_HR | (Drysdale et al., 2020) |
| | | | | | | 696 | CD3-1_LR | (Drysdale et al., 2020) |
| 266 | Cueva Victoria | Spain | 37.6322 | -0.8215 | 266-VICIII1 | 697 | Vic-III-1 | (Budsky et al., 2019; Ros and Llamusí, 2012) |
| | | | | | 266-VICIII3 | 698 | Vic-III-3 | (Budsky et al., 2019; Ros and Llamusí, 2012) |
| | | | | | 266-SR01T | 699 | SR01t | (Budsky et al., 2019; Ros and Llamusí, 2012) |
| 39 | Dongge cave | China | 25.2833 | 108.0833 | 39-D3 | 700 | D3_2005 | (Kelly et al., 2006) |
| | | | | | 39-D4 | 701 | D4_2005_Kelly | (Kelly et al., 2006) |
| 120 | Ejulve cave | Spain | 40.76 | -0.59 | 120-ANDROMEDA | 702 | Andromeda | (Pérez-Mejías et al., 2019) |
| 192 | El Condor cave | Peru | -5.93 | -77.3 | 192-ELCB | 703 | ELC-B_2021 | (Cheng et al., 2021) |
| 115 | Hölloch im Mahdtal | Austria | 47.3781 | 10.1506 | 115-HOL1 | 704 | HOL1 | (Li et al., 2021a) |
| | | | | | 115-HOL22 | 705 | HOL22 | (Li et al., 2021a) |
| 6 | Hulu cave | China | 32.5 | 119.17 | 6-MSL | 706 | MSL_2021 | (Cheng et al., 2021) |
| 10 | Jaraguá cave | Brazil | -21.083 | -56.583 | 10-JAR2 | 707 | JAR2 | (Novello et al., 2019) |
| 24 | Lapa sem fim cave | Brazil | -16.1503 | -44.6281 | 24-LSF13 | 708 | LSF13_2018 | (Stríkis et al., 2018) |

| | | | | | 24-LSF19 | 709 | LSF19 | (Azevedo et al., 2021) |
|---|---|---|---|---|---|---|---|---|
| | | | | | 24-LSF17 | 710 | LSF17 | (Azevedo et al., 2021) |
| | | | | | 24-LSF13 | 711 | LSF13_2021 | (Cheng et al., 2021) |
| 3 | Paraiso cave | Brazil | -4.0667 | -55.45 | 3-PAR27 | 712 | PAR27 | (Cheng et al., 2021) |
| | | | | | 3-PAR15 | 713 | PAR15 | (Cheng et al., 2021) |
| 268 | Pere Noel cave | Belgium | 50 | 5.2 | 268-PN955 | 714 | PN-95-5_2018 | (Allan et al., 2018) |
| | | | | | | | | Allan et al. (2018) |
| 87 | Pindal cave | Spain | 43.4 | -4.53 | 87-CANDELA | 715 | Candela_2023 | (Moreno et al., 2010) |
| | | | | | | 716 | Candela_Base | (Stoll et al., 2022) |
| | | | | | | 717 | Candela_Main | (Stoll et al., 2022) |
| | | | | | | 718 | Candela_L | (Stoll et al., 2022) |
| | | | | | 87-LAURA | 719 | Laura | (Stoll et al., 2022) |
| 295 | Qadisha cave | Lebanon | 34.2439 | 30.0364 | 295-QAD1 | 720 | Qad_1 | (Nehme et al., 2023) |
| | | | | | 295-QAD2 | 721 | Qad_2 | (Nehme et al., 2023) |
| 232 | Rio Secreto cave system | Mexico | 20.59 | -87.13 | 232-RS1 | 722 | RS1 | (Serrato Marks et al., 2021) |
| 219 | Shennong cave | China | 28.71 | 117.26 | 219-SN35 | 723 | SN35 | (Zhang et al., 2021a) |
| | | | | | 219-SN31 | 724 | SN31 | (Zhang et al., 2021a) |
| | | | | | 219-SN29 | 725 | SN29 | (Zhang et al., 2021a) |
| | | | | | 219-SN-COMP | 726 | SN_composite | (Zhang et al., 2021a) |

| | | | | | 219-SN17 | 727 | SN17_2021 | (Zhang et al., 2021a) |
|---|---|---|---|---|---|---|---|---|
| 55 | Sieben Hengste cave | Austria | 46.75 | 7.81 | 55-7H12 | 728 | 7H-12 | (Luetscher et al., 2021) |
| 58 | Spannagel cave | Austria | 47.08 | 11.67 | 58-SPA121 | 729 | SPA121_2021 | (Wendt et al., 2021) |
| | | | | | 58-SPA146 | 730 | SPA146 | (Wendt et al., 2021) |
| | | | | | 58-SPA183 | 731 | SPA183 | (Wendt et al., 2021) |
| | | | | | 58-SPA127 | 732 | SPA127_2023 | (Fohlmeister et al., 2013; Welte et al., 2021) |
| 279 | Staircase cave | South Africa | -34.2071 | 22.0899 | 279-STAIRCASE-COMP | 733 | Staircase_composite | (Braun et al., 2019b) |
| 236 | Toca da Boa Vista | Brazil | -10.1602 | -40.8605 | 236-TBV5 | 734 | TBV5 | (Cheng et al., 2021) |
| | | | | | 236-TBV13 | 735 | TBV13 | (Zhang et al., 2021c) |
| 69 | Xinglong cave | China | 40.5 | 117.5 | 69-XL4 | 736 | XL-4 | (Duan et al., 2019, 2022) |
| 296 | Amir Timur cave | Uzbekistan | 39.4227 | 66.7632 | 296-S124 | 737 | S-12-4 | (Finestone et al., 2022) |
| 94 | Anjohibe | Madagascar | -15.53 | 46.88 | 94-ABC1 | 738 | ABC-1 | (Li et al., 2020) |
| 297 | Bàsura cave | Italy | 44.13 | 8.2 | 297-BA184 | 739 | BA18-4 | (Hu et al., 2022) |
| 298 | Belum cave | India | 15.1 | 78.1 | 298-BLM1 | 740 | BLM-1 | (Band et al., 2022) |
| 299 | Calabrez | Spain | 43.45 | -5.13 | 299-ALICIA | 741 | Alicia | (Stoll et al., 2022) |
| 300 | Careys cave | Australia | -35.07 | 148.66 | 300-CC146 | 742 | CC14-6 | (Scroxton et al., 2021) |
| 301 | Cathedral cave | Australia | -32.617 | 148.94 | 301-WB | 743 | WB | (Markowska et al., 2020) |
| | | | | | 301-WC | 744 | WC | (Markowska et al., 2020) |

| 302 | Crevice cave | South Africa | -34.21 | 22.09 | 302-CREVICE-COMP | 745 | Crevice_composite | (Bar-Matthews et al., 2010) |
|---|---|---|---|---|---|---|---|---|
| 303 | Crystal cave, Australia | Australia | -34.1 | 115 | 303-CRYS1 | 746 | CRY-S1 | (Priestley et al., 2023; Treble et al., 2003) |
| 304 | Cueva Bonita | Mexico | 23 | -99 | 304-CB4 | 747 | CB4 | (Wright et al., 2022) |
| 305 | Cueva Rosa | Spain | 43.4436 | -5.1403 | 305-NEITH | 748 | Neith_2022 | (Stoll et al., 2022) |
| | | | | | | 749 | Neith_2015 | (Stoll et al., 2015) |
| | | | | | 305-ARTEMISAR | 750 | Artemisa_R | (Stoll et al., 2015) |
| | | | | | 305-ANGELINES | 751 | Angelines | (Stoll et al., 2015) |
| 306 | Cuíca cave | Brazil | -11.6822 | -60.6431 | 306-PIM4 | 752 | PIM4 | (Della Libera et al., 2022) |
| | | | | | 306-PIM5 | 753 | PIM5 | (Della Libera et al., 2022) |
| 307 | Efflux cave | South Africa | -33.41 | 22.34 | 307-EFFLUX-COMP | 755 | Efflux_composite | (Braun et al., 2020) |
| | | | | | 307-142843 | 756 | 142843 | (Braun et al., 2020) |
| | | | | | 307-142846 | 757 | 142846 | (Braun et al., 2020) |
| | | | | | 307-142847 | 758 | 142847 | (Braun et al., 2020) |
| | | | | | 307-142848 | 759 | 142848 | (Braun et al., 2020) |
| | | | | | 307-142849 | 760 | 142849 | (Braun et al., 2020) |
| 308 | GD8 | Greenland | 80.3777 | -21.7468 | 308-GD81SLAB1 | 761 | GD8-1 Slab 1 | (Moseley et al., 2021) |
| | | | | | 308-GD81SLAB1ORB | 762 | GD8-1 Slab 1 orb | (Moseley et al., 2021) |
| | | | | | 308-GD81SLAB2 | 763 | GD8-1 Slab 2 | (Moseley et al., 2021) |
| 309 | Goda Mea cave | Ethiopia | 9.49 | 37.66 | 309-GM1 | 764 | GM1 | (Asrat et al., 2018) |

| 310 | Golgotha cave | Australia | -34.083 | 115.05 | 310-GLS1 | 765 | GL-S1 | (Treble et al., 2022) |
|-----|---------------|-----------|---------|--------|----------|-----|-------|-----------------------|
|     |               |           |         |        | 310-GLS2 | 766 | GL-S2 | (Treble et al., 2022) |
|     |               |           |         |        | 310-GLS3 | 767 | GL-S3 | (Treble et al., 2022) |
|     |               |           |         |        | 310-GLS4 | 768 | GL-S4 | (Treble et al., 2022) |
| 311 | Harrie Wood cave | Australia | -35.7 | 148.5 | 311-HWS1 | 769 | HW-S1 | (Tadros et al., 2022, 2016) |
|     |               |           |         |        | 311-HWS2 | 770 | HW-S2 | (Tadros et al., 2022, 2016) |
|     |               |           |         |        | 311-HW38B | 771 | HW_38b | (Tadros et al., 2022, 2016) |
| 312 | Heifeng cave | China | 29.0167 | 107.1833 | 312-HF01 | 772 | HF01 | (Yang et al., 2019) |
| 313 | Herbstlabyrinth cave | Germany | 50.6875 | 8.2058 | 313-HLK2 | 773 | HLK2 | (Waltgenbach et al., 2020) |
|     |               |           |         |        | 313-NG01 | 774 | NG01 | (Waltgenbach et al., 2021) |
|     |               |           |         |        | 313-TV1 | 775 | TV1_2021 | (Waltgenbach et al., 2021) |
|     |               |           |         |        |          | 776 | TV1_2020 | (Waltgenbach et al., 2020) |
| 314 | Herolds Bay cave | South Africa | -34.05 | 22.39 | 314-HEROLDSBAY-COMP | 777 | Herolds_bay_composite | (Braun et al., 2020) |
|     |               |           |         |        | 314-162520 | 778 | 162520 | (Braun et al., 2020) |
|     |               |           |         |        | 314-1625271 | 779 | 162527-1 | (Braun et al., 2020) |
|     |               |           |         |        | 314-162528 | 780 | 162528 | (Braun et al., 2020) |
|     |               |           |         |        | 314-1625272 | 781 | 162527-2 | (Braun et al., 2020) |
| 315 | Huangchao cave | China | 36.6167 | 118.3333 | 315-HC2 | 782 | HC2 | (Tan et al., 2020a) |
| 316 | Hüttenbläserschachthöhle | Germany | 51.3689 | 7.6547 | 316-HBSH1 | 783 | HBSH-1 | (Weber et al., 2021) |

| | | | | | 316-HBSH3 | 784 | HBSH-3 | (Weber et al., 2021) |
|---|---|---|---|---|---|---|---|---|
| | | | | | 316-HBSH4 | 785 | HBSH-4 | (Weber et al., 2021) |
| | | | | | 316-HBSH5 | 786 | HBSH-5 | (Weber et al., 2021) |
| 42 | Ifoulki cave | Morocco | 30.708 | -9.3275 | 42-IFK2 | 787 | IFK2 | (Sha et al., 2021) |
| 317 | Jiangjun cave | China | 22.95 | 104.8167 | 317-JJ0406 | 788 | JJ0406 | (Wassenburg et al., 2021; Liu et al., 2020) |
| | | | | | 317-JJ0403 | 789 | JJ0403 | (Wassenburg et al., 2021; Liu et al., 2020) |
| 318 | Jinfo cave | China | 29.0167 | 107.1792 | 318-J12 | 790 | J12 | (Yang et al., 2019) |
| | | | | | 318-J13 | 791 | J13 | (Yang et al., 2019) |
| 319 | Jiulong cave | China | 27.8 | 113.9 | 319-JL1 | 792 | JL1 | (Zhang et al., 2021b) |
| 320 | Katalekhor | Iran | 35.85 | 48.16 | 320-KT3 | 793 | KT-3 | (Andrews et al., 2020) |
| 100 | Katerloch cave | Austria | 47.0833 | 15.55 | 100-K2 | 794 | K2 | (Honiat et al., 2022) |
| | | | | | 100-K4 | 795 | K4 | (Honiat et al., 2022) |
| 321 | Klang | Thailand | 8.33 | 98.73 | 321-TK7 | 796 | TK7 | (Chawchai et al., 2021) |
| | | | | | 321-TK20 | 797 | TK20 | (Chawchai et al., 2021) |
| | | | | | 321-TK40 | 798 | TK40 | (Chawchai et al., 2021) |
| 322 | Kuna Ba | Iraq | 35.09 | 45.38 | 322-NIR1 | 799 | NIR-1 | (Sinha et al., 2019) |
| | | | | | 322-NIR2 | 800 | NIR-2 | (Sinha et al., 2019) |
| | | | | | 322-NIR-COMP | 801 | NIR_composite | (Sinha et al., 2019) |
| 323 | Kyok-Tash cave | Russia | 51.729 | 85.656 | 323-K4KYOK | 802 | K4_kyok | (Li et al., 2021b) |

| 324 | La Vallina | Spain | 43.41 | -4.8067 | 324-GAEL | 803 | Gael_2022 | (Stoll et al., 2022) |
|---|---|---|---|---|---|---|---|---|
| | | | | | | 804 | Gael_2015 | (Stoll et al., 2015) |
| | | | | | 324-GLORIA | 805 | Gloria | (Stoll et al., 2015) |
| | | | | | 324-GARTH | 806 | Garth | (Stoll et al., 2022) |
| | | | | | 324-GULDA | 807 | Gulda | (Stoll et al., 2022) |
| | | | | | 324-LUNA | 808 | Luna | (Stoll et al., 2022) |
| | | | | | 324-GALIA | 809 | Galia | (Stoll et al., 2022) |
| 221 | La Vierge cave | Mauritius | -19.7572 | 63.3703 | 221-LAVI157 | 810 | LAVI-15-7 | (Li et al., 2020) |
| | | | | | 221-LAVI4 | 811 | LAVI-4_2020 | (Li et al., 2020) |
| 325 | Larga cave | Puerto Rico | 18.32 | -66.8 | 325-PRLA1 | 812 | PR-LA-1 | (Warken et al., 2020) |
| 326 | Linzhu cave | China | 31.5167 | 110.3167 | 326-LZ15 | 813 | LZ15 | (Cheng et al., 2009a) |
| | | | | | 326-LZ36 | 814 | LZ36 | (Cheng et al., 2009a) |
| 327 | Manita peć cave | Croatia | 45.3142 | 15.4754 | 327-MP2 | 815 | MP-2 | (Surić et al., 2021b) |
| | | | | | 327-MP3 | 816 | MP-3 | (Surić et al., 2021b) |
| 328 | Mata Virgem cave | Brazil | -11.62 | -47.49 | 328-MV3 | 817 | MV3 | (Azevedo et al., 2019) |
| 329 | Matupi cave | Democratic Republic of Congo | 1.25 | 29.82 | 329-MAT1 | 818 | MAT1 | (Dupont et al., 2022) |
| | | | | | 329-MAT12 | 819 | MAT12 | (Dupont et al., 2022) |
| | | | | | 329-MAT23 | 820 | MAT23 | (Dupont et al., 2022) |
| 330 | Meravelles cave | Spain | 40.9488 | 0.5127 | 330-MAAT | 821 | Maat | (Pérez-Mejías et al., 2021) |

| 331 | Mizpe Shelagim | Mount Hermon (Levant) | 33.32 | 35.81 | 331-MS-COMP | 822 | MS-composite | (Ayalon et al., 2013) |
|---|---|---|---|---|---|---|---|---|
| | | | | | 331-MS1 | 823 | MS-1 | (Ayalon et al., 2013) |
| | | | | | 331-MS2 | 824 | MS-2 | (Ayalon et al., 2013) |
| | | | | | 331-MS3 | 825 | MS-3 | (Ayalon et al., 2013) |
| 332 | Murada | Spain | 39.9596 | 3.965 | 332-INDIANA | 826 | Indiana | (Torner et al., 2019) |
| 333 | Neotektonik cave | Switzerland | 46.7833 | 8.2667 | 333-M37116A | 827 | M37-1-16A | (Wilcox et al., 2020) |
| | | | | | 333-M37116C | 828 | M37-1-16C | (Wilcox et al., 2020) |
| | | | | | 333-M37123A | 829 | M37-1-23A | (Wilcox et al., 2020) |
| 334 | Nova Grgosova cave | Croatia | 45.8188 | 15.6783 | 334-NG7 | 830 | NG-7 | (Surić et al., 2021a) |
| | | | | | 334-NG3 | 831 | NG-3 | (Surić et al., 2021a) |
| 335 | Ostolo cave | Spain | 43.1878 | -0.2678 | 335-OST2 | 832 | OST2 | (Bernal-Wormull et al., 2021) |
| | | | | | 335-OST1 | 833 | OST1 | (Bernal-Wormull et al., 2021) |
| | | | | | 335-OST3 | 834 | OST3 | (Bernal-Wormull et al., 2021) |
| 336 | Pentadactylos | Cyprus | 35.27 | 33.47 | 336-PENTADACTYLOS1 | 835 | Pentadactylos-1 | (Nehme et al., 2020) |
| 337 | Pir Ghar cave | Iran | 35.23 | 57.42 | 337-PG113 | 836 | PG11-3 | (Carolin et al., 2019a) |
| 338 | Coves del pirata | Spain | 39.5046 | 3.3009 | 338-CONSTANTINE | 837 | Constantine | (Cisneros et al., 2021) |

| 339 | Pozzo Cucù cave | Italy | 40.9 | 17.16 | 339-PC | 838 | PC | (Columbu et al., 2020) |
|-----|-----------------|-------|------|-------|--------|-----|------|------------------------|
| 254 | PP29 | South Africa | -34.2078 | 22.0876 | 254-PP29-COMP | 839 | PP29_composite | (Braun et al., 2019b) |
| 340 | Qujia cave | China | 35.7 | 118.4 | 340-QJ1 | 840 | QJ1 | (Zhao et al., 2021) |
| 341 | Rey Marcos | Guatemala | 15.4277 | -90.2807 | 341-GURM1 | 841 | GU-RM-1 | (Winter et al., 2020) |
| 342 | Sa balma des Quartó cave | Spain | 39.5145 | 3.3059 | 342-SEAN | 842 | Seán | (Cisneros et al., 2021) |
| | | | | | 342-MULTIEIX | 843 | Multieix | (Cisneros et al., 2021) |
| | | | | | 342-CIARA | 844 | Ciara | (Cisneros et al., 2021) |
| | | | | | 342-FENI | 845 | Feni | (Cisneros et al., 2021) |
| 140 | Sanbao cave | China | 31.667 | 110.4333 | 140-SB61 | 846 | SB61 | (Cheng et al., 2009a) |
| 343 | Sant'Angelo cave | Italy | 40.7 | 17.5 | 343-SA1 | 847 | SA1 | (Columbu et al., 2022) |
| 345 | Schratten cave | Switzerland | 46.7833 | 8.2667 | 345-M6733 | 849 | M6-73-3 | (Wilcox et al., 2020) |
| 20 | Secret cave | Borneo | 4.0848 | 114.8503 | 20-SC02 | 850 | SC02_2022 | (Buckingham et al., 2022) |
| 346 | Shijiangjun cave | China | 26.2 | 105.5 | 346-SJJ7 | 851 | SJJ7 | (Chen et al., 2021) |
| 347 | Shizi cave | China | 29.6822 | 106.2881 | 347-QM09 | 852 | QM09 | (Yang et al., 2019) |
| 348 | Sudwala cave | South Africa | -25.37 | 20.7 | 348-SC1 | 853 | SC1 | (Green et al., 2015) |
| 349 | Talisman cave | Kyrgyzstan | 40.39 | 72.35 | 349-F11 | 854 | F11 | (Tan et al., 2021) |
| | | | | | 349-F2TALISMAN | 855 | F2_Talisman | (Tan et al., 2021) |
| 293 | Tham Doun Mai | Laos | 20.75 | 102.65 | 293-TM5 | 856 | TM5 | (Griffiths et al., 2020) |

| | | | | 102.65 | 293-TM4 | 857 | TM4 | (Griffiths et al., 2020) |
|---|---|---|---|---|---|---|---|---|
| | | | | 102.65 | 293-TM11 | 858 | TM11 | (Griffiths et al., 2020) |
| 350 | Toca da Barriguda | Brazil | -10.16 | -40.86 | 350-TBR14 | 859 | TBR14 | (Wendt et al., 2019) |
| | | | | | 350-TBR1013 | 860 | TBR10-13 | (Cheng et al., 2021) |
| 351 | Trapiá cave | Brazil | -5.6 | -37.7 | 351-TRA7 | 861 | TRA7 | (Utida et al., 2020) |
| 352 | War Eagle | United States of America | 34.67 | -86.05 | 352-PPNDA | 862 | PPnda | (Medina-Elizalde et al., 2022) |
| 250 | Wuya cave | China | 33.82 | 105.43 | 250-WY12 | 863 | WY12 | (Tan et al., 2020b) |
| | | | | | 250-WY13 | 864 | WY13 | (Tan et al., 2020b) |
| | | | | | 250-WY14 | 865 | WY14 | (Tan et al., 2020b) |
| | | | | | 250-WY56 | 866 | WY56 | (Tan et al., 2020b) |
| 353 | Wintimdouine | Morocco | 30.77 | -9.49 | 353-WIN1 | 867 | WIN1 | (Sha et al., 2019) |
| | | | | | 353-WIN2 | 868 | WIN2 | (Sha et al., 2021, 2019) |
| | | | | | 353-WIN3 | 869 | WIN3 | (Sha et al., 2021, 2019) |
| | | | | | 353-WIN-COMP | 870 | WIN_composite | (Sha et al., 2019) |
| 354 | Wulu cave | China | 26.05 | 105.0833 | 354-WULU30 | 871 | Wulu-30 | (Cheng et al., 2021; Liu et al., 2018) |
| | | | | | 354-WU88 | 872 | Wu88 | (Liu et al., 2023, p.202) |
| | | | | | 354-WU37 | 873 | Wu37 | (Zhao et al., 2020) |
| 355 | Xiniu cave | China | 31.5167 | 110.57 | 355-XN2 | 874 | XN2 | (Zhao et al., 2021) |

| | | | | | 110.57 | 355-XN15 | 875 | XN15 | (Zhao et al., 2021) |
|---|---|---|---|---|---|---|---|---|---|
| | | | | | 110.57 | 355-XN-COMP | 876 | XN_composite | (Zhao et al., 2021) |
| 5 | Yangkou cave | China | 29.0333 | 107.1833 | 5-JFYK2 | 877 | JFYK2 | (Zhang et al., 2021b) |
| 356 | Yangzi cave | China | 29.783 | 107.783 | 356-YZ1 | 878 | YZ1 | (Wu et al., 2020) |
| 357 | Yonderup cave | Australia | -31.547 | 115.69 | 357-YDS2 | 879 | YD-S2 | (McDonough et al., 2022; Nagra et al., 2017, 2016) |
| 358 | Zhangjia cave | China | 32.5833 | 105.0833 | 358-ZJD171 | 880 | ZJD171 | (Cheng et al., 2021) |
| 359 | Zoolithen cave | Germany | 49.7793 | 11.2829 | 359-ZOOREZ1 | 881 | Zoo-rez-1 | (Riechelmann et al., 2019, 2020) |
| | | | | | 359-ZOOREZ2 | 882 | Zoo-rez-2 | (Riechelmann et al., 2019, 2020) |
| 360 | Bigonda cave | Italy | 46.018 | 11.581 | 360-BG2 | 883 | BG2 | (Johnston et al., 2021) |
| | | | | | 360-BG4 | 884 | BG4 | (Johnston et al., 2021) |
| 117 | Bunker cave | Germany | 51.3675 | 7.6647 | 117-BU1 | 885 | Bu1_2021 | (Waltgenbach et al., 2021) |
| | | | | | 117-BU4 | 886 | Bu4_2021 | (Waltgenbach et al., 2021, 2020) |
| 361 | Kocain cave | Turkey | 37.2325 | 30.7117 | 361-KO1 | 887 | Ko-1 | (Jacobson et al., 2021) |
| 362 | Labyrinth cave | Australia | -34.3 | 115.1 | 362-LABS1 | 888 | LAB-S1 | (Nagra et al., 2017; Priestley et al., 2023) |
| 363 | Lake Shasta cave | United States of America | 40.804 | -122.304 | 363-LSC2 | 889 | LSC2 | (Oster et al., 2020) |

| | | | | | 363-LSC3 | 890 | LSC3 | (Oster et al., 2020) |
|---|---|---|---|---|---|---|---|---|
| 12 | Mawmluh cave | India | 25.2622 | 91.8817 | 12-MAW3 | 891 | MAW-3 | (Magiera et al., 2019) |
| 364 | Naharon | Mexico | 20.18 | -87.54 | 364-NAH14 | 892 | NAH14 | (Warken et al., 2021) |
| 365 | Pot au Feu | Spain | 42.52 | 0.24 | 365-JUD | 893 | JUD | (Torner et al., 2019) |
| 366 | Savi cave | Italy | 45.6167 | 13.8833 | 366-SV1 | 894 | SV1 | (Belli et al., 2013, 2017) |
| | | | | | 366-SV7 | 895 | SV7 | (Belli et al., 2013, 2017) |
| 205 | São Bernardo cave | Brazil | -13.81 | -46.35 | 205-SBE3 | 896 | SBE3_2021 | (Novello et al., 2018, 2021) |
| 225 | Chiflonkhakha cave | Bolivia | -18.1222 | -65.7739 | 225-BOTO1 | 897 | Boto 1_2021 | (Apaéstegui et al., 2018; Novello et al., 2021) |
| | | | | | 225-BOTO3 | 898 | Boto 3_2021 | (Apaéstegui et al., 2018; Novello et al., 2021) |
| | | | | | 225-BOTO7 | 899 | Boto 7_2021 | (Apaéstegui et al., 2018; Novello et al., 2021) |
| 54 | Sahiya cave | India | 30.6 | 77.8667 | 54-SAHA | 900 | SAH-A_2023 | (Sinha et al., 2015) |
| | | | | | 54-SAHB | 901 | SAH-B_2023 | (Sinha et al., 2015) |
| | | | | | 54-SAHAB-COMP | 902 | SAH-AB_2023 | (Sinha et al., 2015) |
| 344 | Sarma cave | Abkhazia (Caucasus) | -25.37 | 20.7 | 344-SAR121 | 848 | SAR-12-1 | (Wolf et al., 2024) |

Table 7: New entities added to SISALv3.

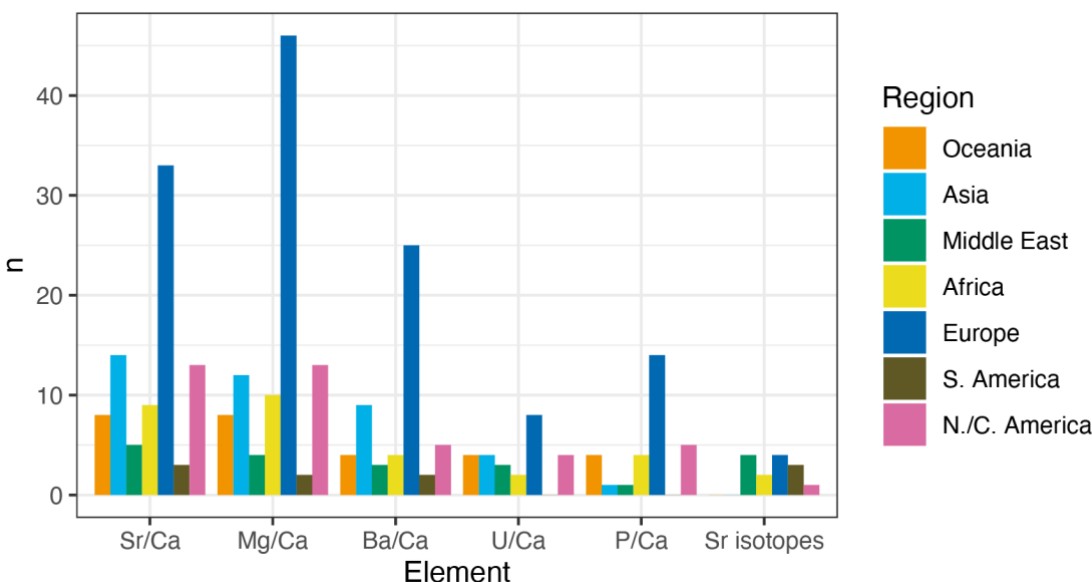

*Figure 3: Trace element ratios and Sr isotope records included in SISALv3 by region. Abbreviations: S. America - South America, N./C. America - North and Central America.*

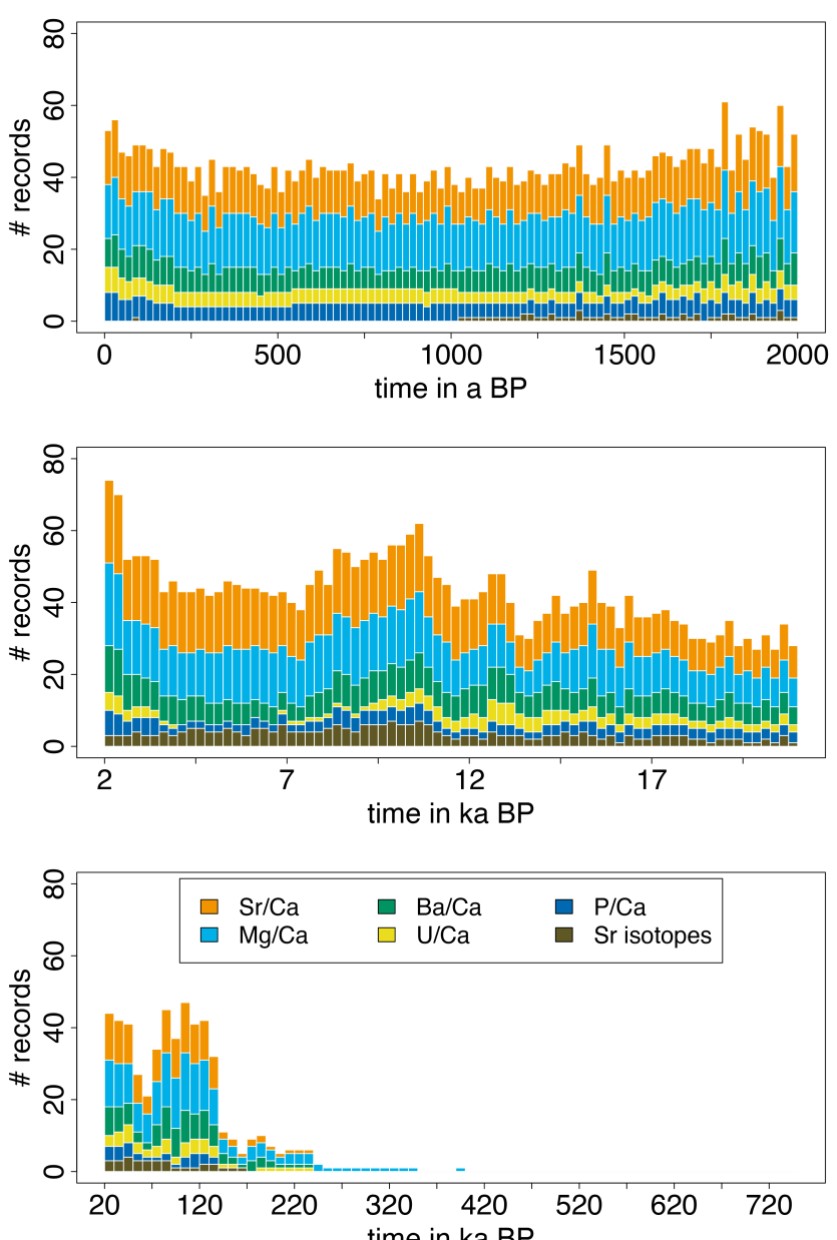

*Figure 4: Temporal coverage of the trace element and Sr-isotope records in SISALv3 by region. Entities with multiple trace elements were counted multiple times. Bin sizes: 0-2,000 a (years) BP (top panel) - 20 years; 2,000-21,000 a BP (middle panel) - 250 years; 21,000-750,000 a BP (bottom panel) - 10,000 years.*

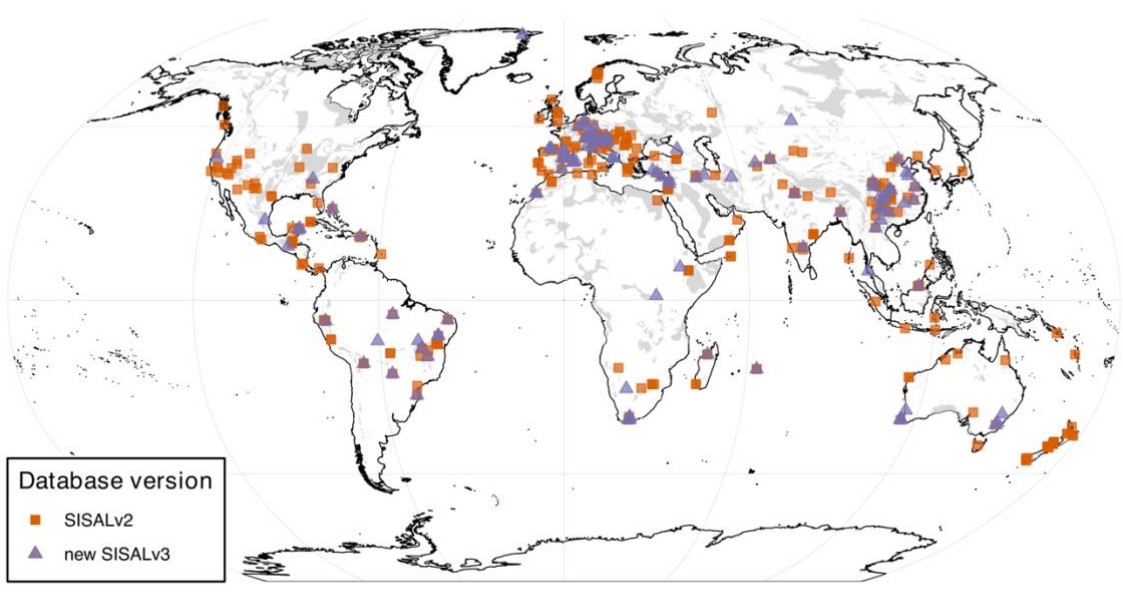

*Figure 5: Global map of δ¹⁸O records included in SISAL v2 and v3. The shaded background shows the global karst distribution extracted from the World Karst Aquifer Map (WOKAM, Goldscheider et al., 2020).*

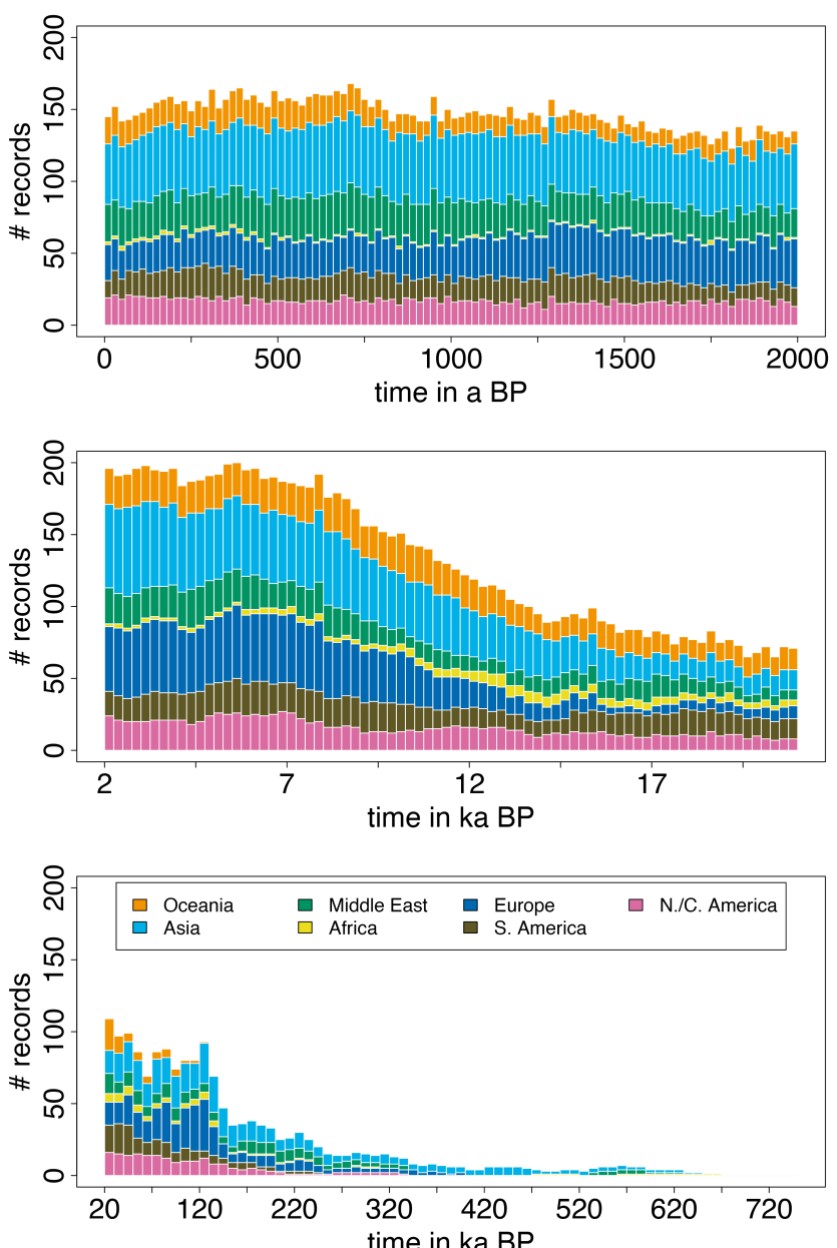

*Figure 6: Temporal coverage of the $\delta^{18}O$ records in SISALv3 by region. Bin sizes: 0-2,000 a BP (top panel) - 20 years; 2,000-21,000 a BP (middle panel) - 250 years; 21,000-750,000 a BP (bottom panel) - 10,000 years.*

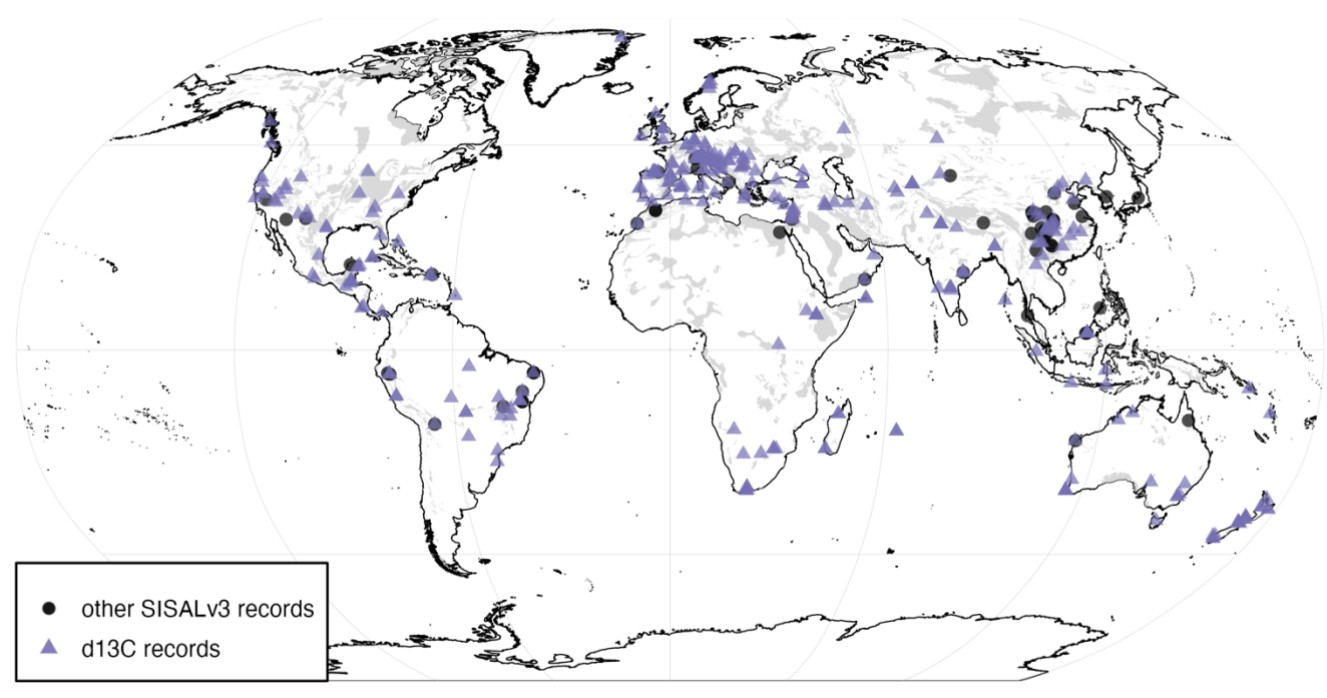

*Figure 7: Map of available δ¹³C records in SISALv3 compared to all records in the database. The shaded background shows the global karst distribution extracted from the World Karst Aquifer Map (WOKAM, Goldscheider et al., 2020).*

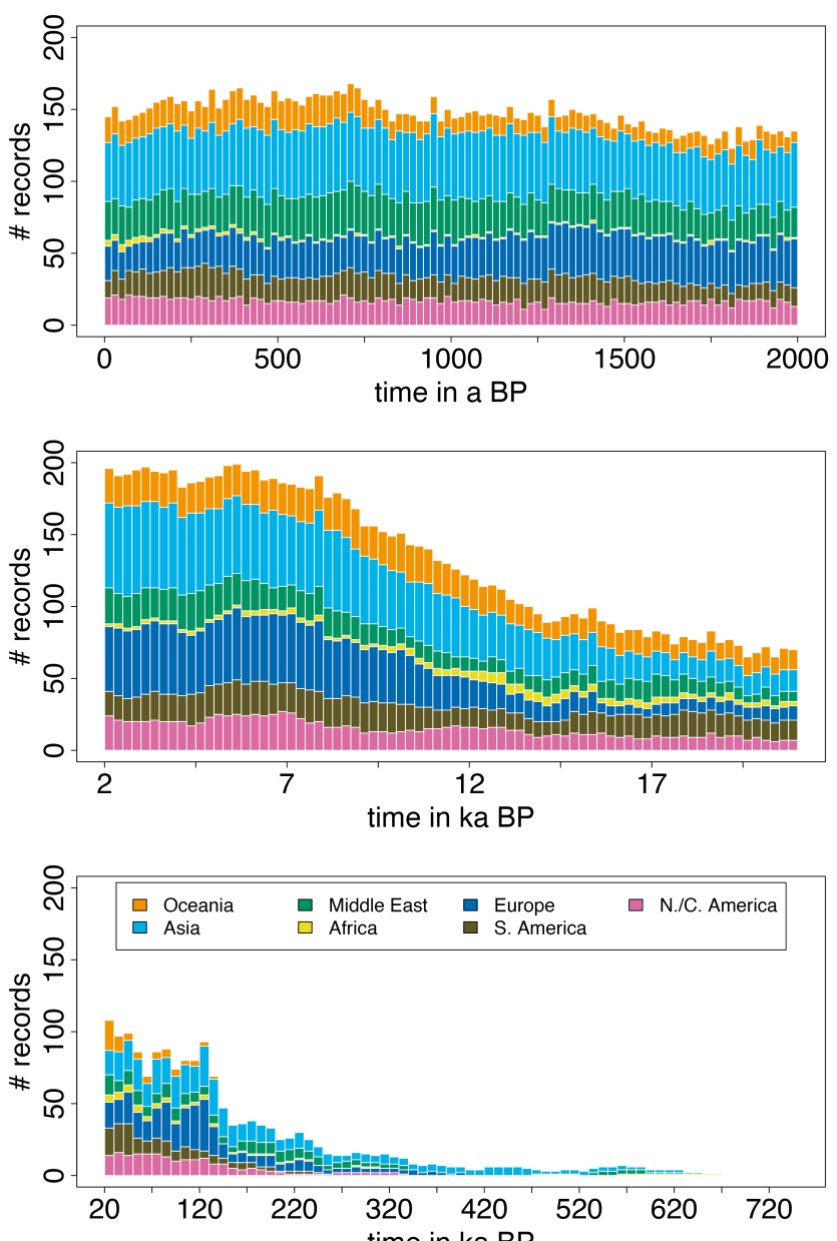

*Figure 8: Temporal coverage of the δ¹³C records in SISALv3 by region. Bin sizes: 0-2,000 a BP (top panel) - 20 years; 2,000-21,000 a BP (middle panel) - 250 years; 21,000-750,000 a BP (bottom panel) - 10,000 years.*

## 4.3 Vegetation and land cover metadata

Interpretation of the site-to-site variability in speleothem data sensitive to vegetation changes is facilitated by providing information on *vegetation_type* and *land_use*. The dropdown list for these fields includes options typically used in speleothem publications. Additional information provided by the authors (e.g. species names) has been added to the Notes table. About 40% of the database entries lack the author-reported information on land cover (Figure 9c). Satellite-derived land cover classifications provide information for many more sites (unknown: 1.7%; Figure 9d). Forested sites (evergreen, deciduous, and mixed) comprise ~56.5% (Figure 9d), shrub- and grassland makes up 25.1% and this dataset also denotes sites that are affected by anthropogenic land use (managed vegetation, agriculture, urban), which make up 13%.

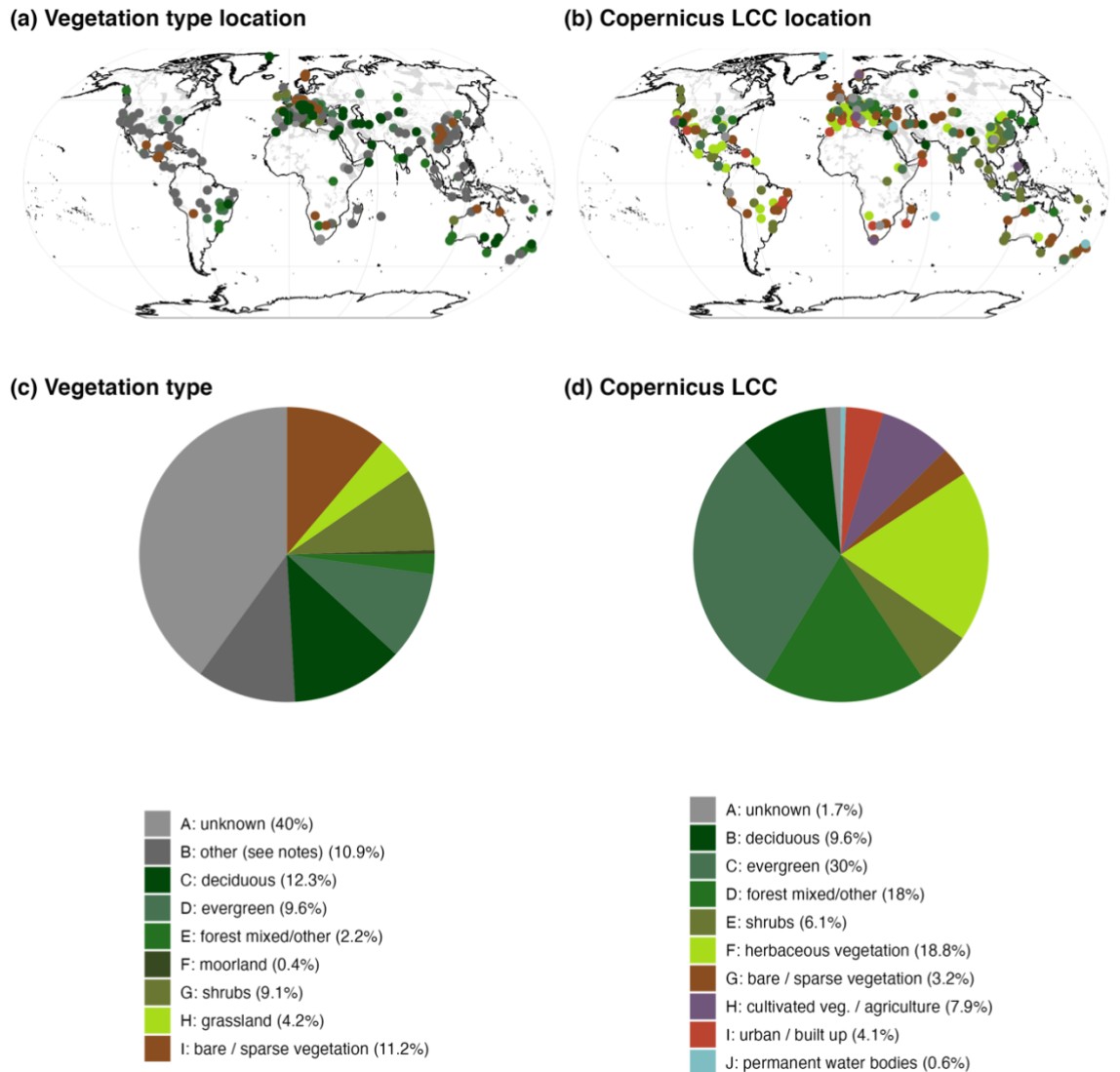

*Figure 9: (a) – vegetation description from the original publications or provided by authors and (b) – land cover categories extracted from the Copernicus LCC database (Buchhorn et al., 2021, 2020) with a radius of 250 m around the cave sites. (c) - pie chart showing the relative proportions of vegetation types as reported by authors. (d) - pie chart showing the relative proportions of land cover types as extracted from the Copernicus LCC database. Background shading in the map shows the*

310 *global karst distribution extracted from the World Karst Aquifer Map (WOKAM, (Goldscheider et al., 2020). To allow comparison between the two datasets, the Copernicus LCC vegetation data was grouped into broader categories, e.g., "deciduous" includes all closed and open broad-leaf and needle forest marked as deciduous. The entries in the database are more detailed.*

## 5 Recommendations for use

The SISALv3 database is a standardized, quality checked dataset that allows regional to global assessments of spatial and temporal trends in multiple environmental proxies from speleothem records. The addition of trace element data at stable isotope equivalent depths to the database together with machine-readable metadata fields allow examination of hydroclimatic controls on speleothem trace element distribution. Metadata fields, including distance from coast (*latitude, longitude, elevation*), lithology (*geology*, *wokam*), and land cover (*cover_type*, *cover_thickness, vegetation_type, land_use, copernicus_lcc*), allow identification of the primary controls on trace elements. We recommend using multiple cover fields together, based on the analysis type and scope (e.g., time interval considered) since they provide complementary information. Anthropogenic and natural changes in the cover parameters over time need to be considered, and this applies particularly for the cover fields "*vegetation_type*", "*land_use*" and "*copernicus_lcc*", which in most cases may only be applicable for very recent speleothem growth.

Where trace elements are measured on aliquots of the same powder as stable isotopes, the sample-to-sample variability in depth-time space is minimal. Where samples for stable isotopes and trace elements have been drilled at different times or *in situ* methods have been used for trace element measurements, there may be depth-time variability that may impact results. Extensive metadata on sampling and measurement methods, as well as the original high resolution *in-situ* measurements against depth, are provided in the database and linked repository and should be used to check for such impacts. Measurements may also be sensitive to stalagmite petrography; image scans have been provided in the linked repository so the user can evaluate whether this is important for interpretation of the record.

### 5.1 Code and data availability

The database is available in CSV and SQL format in a repository at https://doi.org/10.5287/ora-2nanwp4rk (Kaushal et al., 2024). This dataset is licensed by the rights holder(s) under a Creative Commons Attribution 4.0 International License: https://creativecommons.org/licenses/by/4.0/. Apart from the workbook used to submit data to the SISAL database and the codes for automatic quality checking, the repository contains additional standardisation sheets (coordinate conversion, grams to moles conversion for trace elements, and atomic activity calculator for U-series data). Moreover, the repository contains all submitted cave maps and entity images in separate zip folders, as well as copyright information for the individual images and an entity scan "wishlist" which details best practices for entity scan images. Standardized trace element datafiles are included separately with their metadata (see section 2), and the codes needed to connect and use the database (described in the Readme file).

The codes for standardisation and downsampling of trace element and Sr-isotope records are available at zenodo 10.5281/zenodo.8234066 (Skiba, 2023); licensed by the right holder(s) under Creative Commons Attribution 4.0 International).

The database contains both the original age model for individual entities and a standardized age modelling ensemble. The original age model often takes account of site- and sample-specific conditions; the standardized age model ensemble allows for robust assessment of age uncertainties and sensitivity testing (Comas-Bru et al., 2020). All codes for constructing the age

model ensembles using linear interpolation, linear regression, Bchron, Bacon, copRa, and StalAge can be found at https://github.com/paleovar/SISAL.AM (last access: 23 July 2020; codes licensed by the right holder(s) under a GPL-3 license). These codes are licensed by the right holder(s) under a Creative Commons Attribution 4.0 International. All age model ensembles are available at https://zenodo.org/records/10726619 (Rehfeld and Bühler, 2024). These codes are licensed by the right holder(s) under a Creative Commons Attribution 4.0 International.

The SISALv3 database, like its predecessors, lists the original references, and users are encouraged to consult original authors for interpretative details. The 'SISAL webApp' (http://geochem.hu/SISAL_webApp ; Hatvani et al., 2024) has been updated to provide easy-to-use front-end interface in exploring the latest SISALv3 database. It now allows for querying on various data and metadata fields such as stable isotope records and trace element proxies.

**5.2 How to cite the database**

The SISALv3 database is a community driven effort to synthesize, standardize and make speleothem data to the wider paleoclimate community. In agreement with the FAIR principles for scientific data management and stewardship, the database itself should be cited (available at https://doi.org/10.5287/ora-2nanwp4rk; Kaushal et al., 2024), together with this publication (and previous version publications). If individual records are extracted from the database, the original publications should also

be listed. More details on Terms of Use are provided in the repository (https://doi.org/10.5287/ora-2nanwp4rk; Kaushal et al., 2024).

**Author contributions**

NK coordinated this project. NK, FL and MW designed the new version of the database. KR and JB ran the SISAL standardized age-depth models for new entities. Downsampling of trace element records to stable isotope resolution was performed by VS

and MR. Standardization of trace element datafiles was done by YB and NK. Reworking and additions to the metadata fields were done by KB and KA. JGS and NK collected citations, copyright information and license terms for the cave maps and speleothem images. Regional data collection and screening was coordinated by VA, JLB, SC, AC, LE, JH, IGH, ZK, AK, KK, MK, BM, SMA, CN, VFN, CPM, JR, NS, NiS, CT, BHT, SW, AW, HZ. Quality control of the submitted datasets was performed by MW, FL, and NK, with additional code provided by JF. Figures 1 and 2 were created by FL, Figures 3-9 were

created by JB. All authors listed as "Data contributors" provided data for this version of the database or helped to complete existing data entries. FL wrote the paper with input from NK, JB, KR, AB, PT, SPH, and all authors contributed to the final version.

**Team list**

The following SISAL working group members contributed with either data or age-modelling advice to SISALv3: Asfawossen
Asrat (Department of Mining and Geological Engineering, Botswana International University of Science and Technology,
Private Bag 16, Palapye, Botswana), Charlotte Honiat (Institute of Geology, University of Innsbruck, Innrain 52, Innsbruck,
Austria), Dana Felicitas Christine Riechelmann (Institute for Geosciences, Johannes Gutenberg University Mainz, Johann-
Joachim-Becher-Weg 21, 55128 Mainz, Germany), Denis Scholz (Institute for Geosciences, Johannes Gutenberg University
Mainz, Johann-Joachim-Becher-Weg 21, 55128 Mainz, Germany), Dianbing Liu (School of Geography, Nanjing Normal
University, Nanjing 210023, China), Dominik Fleitmann (Department of Environmental Sciences, University of Basel,
Bernoullistrasse 32 4056 Basel, Switzerland), Dominik Hennhoefer (Department of Earth Sciences, Khalifa University (SAN
Campus), Abu Dhabi, 127788, United Arab Emirates), Ezgi Ünal İmer (Geological Engineering Department, Middle East
Technical University, 06800 Çankaya, Ankara, Türkiye), Gina E. Moseley (Institute of Geology, University of Innsbruck,
Innrain 52, 6020 Innsbruck, Austria), Giselle Utida (Institute of Geosciences, University of São Paulo, 05508-080, Brazil),
Hai Cheng (Institute of Global Environmental Change, Xi'an Jiaotong University, China), Helen Green (The University of
Melbourne, Parkville VIC 3010, Australia), Hsun-Ming Hu (High-Precision Mass Spectrometry and Environment Change
Laboratory (HISPEC), Department of Geosciences, National Taiwan University, Taipei 10617 Taiwan), James Apaéstegui
(Instituto Geofísico del Perú, Lima, 15012, Peru), Jan Esper (Department of Geography, Johannes Gutenberg University,
Becherweg 21, 55099 Mainz, Germany), Jasper A. Wassenburg (1. Center for Climate Physics, Institute for Basic Science,
Busan 46241, Republic of Korea 2. Pusan National University, Busan, Republic of Korea), Jeronimo Aviles Olguin (Museo
del Desierto. Blvd. Carlos Abedrop Dávila 3745, Nuevo Centro Metropolitano de Saltillo, 25022 Saltillo, Coah. Mexico),
Jessica Leigh Oster (Department of Earth and Environmental Sciences, Vanderbilt University, Nashville, TN 37240, USA),
Jesús M. Pajón Morejón (National Museum of Natural History of Cuba, Department of Paleogeography and Paleobiology,
Obispo 61, Plaza de Armas, Habana Vieja, CP 10 100, La Habana, Cuba), Judit Torner (CRG Marine Geosciences, Facultat
de Ciències de la Terra, Universitat de Barcelona, Barcelona, 08028, Spain), Kathleen A Wendt (College of Earth, Ocean, and
Atmospheric Sciences, Oregon State University, Corvallis, Oregon 97331, USA), Liangcheng Tan (State Key Laboratory of
Loess and Quaternary Geology, Institute of Earth Environment, Chinese Academy of Sciences, Xi'an 710061, China), Lijuan
Sha (Institute of Global Environmental Change, Xi'an Jiaotong University, Xi'an 710049, China), Liza Kathleen McDonough
(ANSTO, New Illawarra Road, Lucas Heights, NSW 2234, Australia), Maša Surić (Department of Geography, University of
Zadar, Ul. dr. F. Tuđmana 24 i, Zadar 23000, Croatia), Matthew J. Jacobson (Division of Agrarian History, Department of
Urban and Rural Development, Swedish University of Agricultural Sciences, Uppsala, 756 51, Sweden), Mercè Cisneros (1.
GRC Geociències Marines, Departament de Dinàmica de la Terra i de l'Oceà, Facultat de Ciències de la Terra, Universitat de
Barcelona. c/ Martí i Franqués s/n, 08028 Barcelona, Spain 2. Centre en Canvi Climàtic, Department de Geografia, Facultat
de Turisme i Geografia, Universitat Rovira i Virgili, c/ Joanot Martorell 15, 43480, Vila-seca, Tarragona, Spain), Michael L.
Griffiths (Department of Environmental Science, William Paterson University, Wayne NJ, 07739, USA), Michael Weber

(Institute for Geosciences, Johannes Gutenberg University Mainz, J.-J.-Becher-Weg 21, 55128 Mainz, Germany), Nick Scroxton (Irish Climate and Analysis Research UnitS (ICARUS), Department of Geography, Maynooth University, Maynooth, Co. Kildare, Ireland), Paul S. Wilcox (Institute of Geology, University of Innsbruck, Innrain 52, 6020 Innsbruck, Austria), R. Lawrence Edwards (Department of Earth and Environmental Sciences, University of Minnesota, Minneapolis, MN 55455, USA), Romina Belli (Proteomics and Mass Spectrometry Core Facility, Department of Cellular, Computational and Integrative Biology (DeCIBIO), University of Trento, Via Sommarive 9, 38123 Trento, Italy), Sebastian F.M. Breitenbach (Department of Geography and Environmental Sciences, Northumbria, Newcastle upon, Tyne NE1 8ST, UK), Shraddha T Band (National Taiwan University, Address: Institute of Oceanography, National Taiwan University  No.1, Sec. 4, Roosevelt Road, Taipei 106 ,Taiwan), Simon Dominik Steidle (Institute of Geology, University of Innsbruck, Innrain 52, 6020 Innsbruck, Austria), Stacy Anne Carolin (Department of Earth Sciences, University of Cambridge, Downing Street, Cambridge, CB23 8AD, UK), Vanessa E. Johnston (Karst Research Institute ZRC SAZU, Titov trg 2, 6230 Postojna, Slovenia), Wuhui Duan (1. Key Laboratory of Cenozoic Geology and Environment, Institute of Geology and Geophysics, Chinese Academy of Sciences, Beijing,100029, China 2. CAS Center for Excellence in Life and Paleoenvironment, Beijing, 100044 China).

**Competing interests**

The authors declare that they have no conflict of interest.

**Funding**

This study was conducted by SISAL (Speleothem Isotopes Synthesis and Analysis), a working group of the Past Global Changes (PAGES) project. In this framework, we received financial support from the Swiss Academy of Sciences and the Chinese Academy of Sciences. The design and construction of the SISALv3 database were supported financially by a PAGES Data Stewardship Scholarship to F. Lechleitner and N. Kaushal (DSS-108). We also acknowledge funding by PAGES, the Minerva Stiftung (grant 3063000253), and the Institute of Earth Sciences at the Hebrew University Jerusalem (Israel) for supporting the organisation of a workshop to kickstart the initiative.

**Acknowledgement**

We thank all SISAL members who contributed their data to this project and were available to provide additional information where necessary. We also acknowledge Ana Moreno, Christoph Spötl, and Laura A. Dupont for specific data contributions to

the database. We thank the editorial staff at ESSD and reviewers Ewan Gowan, Christopher Hancock, Sang Chen, and an anonymous referee for their supportive and critical feedback to this manuscript.

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
