# Peer review of "SISALv3: A global speleothem stable isotope and trace element database"

_Earth System Science Data, 2023_

## Author Response (AR1)

**Referee Comment file**

**Author responses are in blue below the referee comments.**

**Referee Comment 1:**

Evan J. Gowan

Citation: https://doi.org/10.5194/essd-2023-364-RC1

Review of: SISALv3: A global speleothem stable isotope and trace element database by Kaushal et al

Kaushal et al present an update of the SISAL database, adding many new records and incorporating new isotope proxies including Mg/Ca, Sr/Ca, Ba/Ca, U/Ca, P/Ca and Sr. These additions necessitated a slight restructuring of the database, which is available as a MySQL database, and csv files. In the paper, they have detailed all of the data fields in the database, as well as highlighting the locations of the data and the time periods they cover. In addition to the speleothem data, they also include land cover and vegetation data, which is useful for interpreting the data.

The editor, Dr. Wang, asked me to provide a more formal review after my previous comment. This is the third major iteration of this database, so at this point it is quite mature and useful. As mentioned before, it is a great credit to the speleothem community that updates to this database happen frequently, and I look forward to the eventual version 4. I have found it useful for my own research, and I am sure many others do as well. After a more detailed read of the manuscript, I think expanding on the following points would be useful to add to the manuscript to aid those who want to use the database (in addition to the previous point I made of including a table of the new records).

Thank you for your kind remarks and helpful suggestions. We are glad that you find the database useful.

- With all the new elemental proxies added, it would be useful to include a table describing what climatic variables the proxies are representing. This would strengthen the stated goal of the paper to assessing hydroclimate variability
  - We understand that this version of the database includes 7 proxies which is a large number! As you suggest, it would be very useful to have a table with climatic interpretations given against these proxies. However, the big strength of these proxies is that they provide robust climatic interpretations through a *multi-proxy* approach and we are wary that a table may be an oversimplification of proxy interpretations. In addition, different proxies can be used together in different karst and climatic situations to provide climate information. As in the past with oxygen isotopes (e.g.Quaternary Special Issue ISSN 2571-550X: Speleothem records and climate, Harrison and Comas-Bru, 2018) and carbon isotopes (e.g. Fohlmeister et al, 2020), the SISAL Working Group is already working on further publications with the new additional proxies to provide more nuanced climatic interpretations. This is more time-consuming than a table, but we think ultimately provides a more accurate interpretation of the proxies in the database. At the moment in the manuscript, we have provided a table with a number of published papers as examples of how these proxies have been used for paleoclimate interpretations. We must emphasize that the list is not-exhaustive, the interpretations are time-scale dependent, and in most cases, multi-proxy approaches are necessary.
  - The revised manuscript now includes interpretations in **Table 1** and the following text in **line sections 120-125**, '*The strength of these proxies is that they provide robust climatic and environmental information via a multi-proxy approach that will need to*

*be tailored for different karst and climatic settings (Table 1). The SISAL Working Group is currently working on projects with the new additional proxies to explore and gain more detailed insights. We provide examples of proxy interpretations with linked references, but we must emphasize that this list is not exhaustive, the interpretations are time-scale dependent, and in most cases, multi-proxy approaches are necessary (Table 1).'*

- I believe there has been an online interactive version of the database made available since the publication of SISALv2. It might be a good idea to have a short section explaining how that can be used to access the data.
  - Thank you for the suggestion. The SISAL webApp has been updated to work with the SISALv3 database and will be made available to the public upon the acceptance of the present database paper.
  - The revised manuscript now contains the following text in **line section 355**, *'The SISALv3 database, like its predecessors, lists the original references, and users are encouraged to consult original authors for interpretative details. The 'SISAL webApp' (http://geochem.hu/SISAL_webApp ; Hatvani et al., 2023) has been updated to provide easy-to-use front-end interface in exploring the latest SISALv3 database. It now allows for querying on various data and metadata fields such as stable isotope records and trace element proxies.'*

- The first version of SISAL came out a few years ago. Having a short section showing some example applications of the database would highlight why this database is so important.

  - Some example applications of the database have been listed in **line sections 100-105**. However, adding more text is beyond the scope of such a database format paper.

**Referee Comment 2:**

Chris Hancock

Citation: https://doi.org/10.5194/essd-2023-364-RC2

Kaushal et al. describe the newest version of the SISAL database (SISALv3). I have found previous versions of the SISAL database to be incredibly valuable in my research, and the numerous of publications related to SISAL data (as listed in the introduction) also speaks to this fact. The compilation and standardization of so many records is impressive, and this paper builds on that previous work.

Overall, the manuscript is well written and provides extensive detail about the changes between SISALv3 compared to SISALv2. The most significant update is the addition of trace element and Sr isotope records. As the authors note, the inclusion of trace element data will help researchers understand hydroclimate and hydrological processes. New stable isotope data are also provided in the updated database. Minor changes to the database structure and metadata are also discussed. These include the addition of land cover and vegetation metadata fields. The changes represent a significant advance and make SISALv3 a valuable contribution to the field.

Thank you for your kind words and for your thorough review of the manuscript and the database. We appreciate your time and found the comments very helpful. We have responded to your comments below.

Below is a list of minor comments about the manuscript which I believe will help clarify the text and improve the usability of the SISAL database:

1.  Line 69. SISAL does not include climate interpretations. However, both the abstract and introduction justify the need for adding trace element data to the database as "hydroclimate-sensitive geochemical proxies". As SISAL data is used by a wide-variety of researchers, including those with limited speleothem expertise, expert-defined climate interpretations would be valuable to prevent misinterpretations.
    -   This has been addressed in our response to Referee Comment 1.

2.  Line 81. Does SISAL indicate which data are at seasonal resolution?
    -   SISAL does not have an explicit field to indicate which data are at seasonal resolution. In some cases, high resolution sampling of stable isotopes and trace elements (particularly laser ablation-based methods for trace element measurements) may suggest seasonal resolution data when looking at the chronology fields. However, given the variable residence time of karst waters, the complexity with which seasonal records maybe preserved in the stalagmites due to seasonal in-cave processes (such as seasonally driven ventilation regimes) and the sampling of the stalagmite growth axes itself, we strongly suggest careful consideration of the original publications before ascribing any seasonality to the records in the database.

3.  Lines 144 and 146. Why were the radius of 250 m and 1000m selected? What is the spatial resolution of these datasets?
    -   The WOKAM data is provided in a multiple polygon vector format so there is no grid, which makes estimating the resolution of the dataset meaningless. Essentially each polygon defines regions of certain karst classification. The vertices for these polygons are unevenly spaced. We assume a 1000 m radius around the cave site for the WOKAM field as a reasonable extent of a karst unit in the field.

- The copernicus LCC product is gridded on a 100 m horizontal resolution. By assuming a radius of 250 m we ensure sufficient pixels are selected around the coordinates of the cave entrance. We assign the value of the most frequent landcover type occurring within this range.

4. Line 208: It should be clear how records from previous versions have been updated. How does SISAL version control and where is this information listed for users to access?
    - Consistent with the previous version of the database, SISALv2, (Comas-Bru et al, 2020), changes made to the fields, pre-defined options and to the data already in the previous version of the database are listed in tables in the manuscript. For SISALv3, changes made to the fields and pre-defined options are listed in **Table 2**, and changes made to the data already in the database are summarized in **Table 3** of the revised manuscript.

5. Line 211: I suggest changing the caption from "…datasets in SISALv3" to "…datasets in SISAL" as there does not appear to be any change in this process from previous versions. More generally, the text of section 3 could be streamlined as there appears to be minimal differences from previous versions other than that there is new data to QC.
    - Thank you for your comment which we are happy to take on board. We have streamlined the text of **section 3**.
    - There is one change in the process adopted for SISALv3 that is different from previous versions which is that the autoQC scripts on the submitted files were run centrally by the SISAL database managers rather than by Regional Coordinators as in previous versions. This overall proved to be a more efficient workflow.

6. Line 215: This table is useful, but it would be more valuable to know which records have been modified. For example, which 4 time series were altered? I see that there is some documentation in the notes.csv, but this is not easily searchable to find this specific information.
    - Table 3 shows that a number of modifications have been made to the database between SISALv2 and SISALv3. Some changes are largely straightforward, i.e.
        - the addition of new sites, new entities, new proxies (i.e. trace elements) or the addition of d13C data to the existing entities
        - the addition of new / missing metadata from publications or by data contributors e.g. geology and rock age.
    - Some changes are corrections made to the database as they have been found while using the data for scientific projects by the SISAL Working Group, these especially relate to values of dating.234_238_activity, dating.234_238_activity_uncertainty, dating.ini_230_232_Th_ratio and dating.ini_230_232_Th_ratio_uncertainty.
    - And lastly, the entities that had d18O / d13C time series altered (n=4). Since this is probably of most interest to database users, these entities have been listed in **line sections 260-265** of the revised manuscript. *'Four $\delta^{18}O$ records previously included in SISALv2 have been modified to correct previous mistakes (Table 4), these are entity_ids 110 (CUR4; Novello et al., 2016), 169 (Dim-E3; Ünal-İmer et al., 2015), 447 (JAR4; Novello et al., 2017), and 573 (Gej-1; Flohr et al., 2017).'* And *'two $\delta^{13}C$ records previously included in SISALv2 have been modified to correct previous mistakes (Table 4), these are entity_ids 169 (Dim-E3; Ünal-İmer et al., 2015), and 573 (Gej-1; Flohr et al., 2017).'*

    -

7. Line 222: What is the 60% number based off?
   - The 60% number is based on a 'masterlist' maintained by the SISAL Working Group where we keep a record of all the published / known records, and how many of these have been entered into the database. A clarifying statement has been added to the manuscript in **line section 240**: *'This corresponds to ~60% of the known published data, based on an assessment by the SISAL WG.'*

8. Line 230: My understanding of Table 1 is that data with trace element higher resolution values would be identifiable by the "trace_elements_metadatafile" column of the entity table. However, when I open this csv, there are no rows with a "yes" value in this column.
   - **Line section 245** in the revised manuscript says that, '*Where the original laser ablation data have been provided by data contributors, these have been made available as *.txt datafiles in the repository (Table 4).'* These records are identifiable by the "trace_elements_**datafile**" field rather than the "trace_elements_**metadatafile**" field. You are correct that there is not a single record that has submitted a "trace_elements_metadatafile" sheet. This sheet had been created during our workshop as a list of 'best practices' metadata that should be made available with trace element data. A number of these metadata fields were later incorporated as 'filterable fields' with dropdown list options and with the decision to incorporate stalagmite section images/scans within the framework of the database itself. Nevertheless, we think that it is important that there is consistent reporting of metadata linked to trace element measurements and therefore this field has been retained. These 'wishlists' have now been included in the repository entry linked to SISALv3 and detailed in the **Repository README file**.

9. Line 250: It is unclear to me why the authors chose to show the increase as percentages rather than as count. A count seems more relevant to describing the actual data.
   - The increases in records by region have been changed to counts. See **Table 6** of the revised manuscript.

10. Line 255: A map of trace element records would be valuable to see.
    - Thank you for your comment. While we agree that it would be nice to have the trace element data on a map, previous attempts at this have shown it to be quite difficult (either too many symbols on one map or too many small maps, either resulting in unintelligible results). This is why we opted for the histogram with geographic distribution of trace element records. The histogram version also provides temporal distribution information in addition to spatial distribution information.

11. Line 268 (Figure 7): It is unclear to me why Figure 5 and Figure 7 are so similar but have different legends. Couldn't both maps shows SISALv2 data in orange, new data in purple, and other records in black? It would also be interesting to see which sites had alterations (as listed in table 3).
    - Suitable colour and marker modifications have been made to both maps i.e. **Figure 5** and **Figure 7** of the revised manuscript.
    - As comment 6 explains in more detail, marking which sites have alterations is not straightforward. The important alterations in records have been detailed in the text instead.

12. Line 278: Do "notes" include any information about past vegetation and land use changes or is everything based on modern?
    - 'Notes' mainly include information on modern vegetation and land use as given by the data contributors. In a few cases, the data contributors may have specified that the forest growth for example is a modern planted one.

13. Line 285: The Figure 9 captain is missing a description of panels C & D. The transparency also makes it difficult to see which vegetation type corresponds to each dot.
    - Thank you for pointing this out. We have added descriptions for **Figure 9** panels 'c' and 'd' to the caption and have reduced the transparency in the figure. We hope this improves readability.

14. Line 295: It is unclear to me how "using all cover fields together" would be manageable in a programmatic big data analysis. Could the authors provide context to how this may be done?
    - By 'all cover fields' we mean all fields that can be used to provide information about the land cover at the cave site including: geology, wokam, vegetation_type, land_use, copernicus_lcc and cover_type. A selection would have to be made for programmatic big analysis though. For example, it is unlikely that the geology and wokam fields, and perhaps the cover_type field would have changed over the growth period of the speleothem. These fields provide additional information to interpret speleothem mineralogy and sources for speleothem trace elements. For example, a speleothem retrieved from a drip site with dolomite geology from a continuous carbonate region is more likely to have aragonite mineralogy, low (and less variable) stalagmite Mg/Ca ratios. Vegetation_type, land_use and copernicus_lcc are applicable only to very recent speleothem growth. For the same speleothem, the vegetation_type, land_use and copernicus_lcc can be used to understand baseline d13C values which are useful for understanding the impact of Prior Carbonate Precipitation on d13C values.
    - We have updated the manuscript to reflect this with the following: *"We recommend using multiple cover fields together, based on the analysis type and scope (e.g., time interval considered) since they provide complementary information. Anthropogenic and natural changes in the cover parameters over time need to be considered, and this applies particularly for the cover fields "vegetation_type", "land_use" and "copernicus_lcc", which in most cases may only be applicable for very recent speleothem growth."* See **line section 320** of the revised manuscript.

15. Line 334: Will the Rehfeld et al. (2020) zenodo link be updated to include age model ensembles for new sites?
    - Yes, the Rehfeld et al Zenodo link will include age-depth ensembles for the new entities as well. The updated DOI has been added to the revised manuscript.

16. Line 333: To encourage proper credit of data creators, the text "If individual records are extracted" should be revised to clearly state than any analysis of SISAL records should credit the original publications. The current text seems to suggest that data credit is only needed if a single record is extracted. I would also like to see a list of references for new data in SISALv3.

    - A list of references for new data in SISALv3 are listed in **Table 7** of the revised manuscript.
    - Some studies based on the database may well include upwards of 10's of individual publications reaching the publisher's limit of number of individual references that can be added to a single manuscript. We therefore feel that our compromise of necessarily citing the database publications, and in addition, citing individual original data contributing publications where possible, is a practical compromise.
    - All data contributors / data generating groups were contacted when publishing this manuscript, and as long as they agreed, they are co-authors on

this database paper. The same method has been followed for all previous SISAL WG database publications.  In some cases, they preferred to be listed in the acknowledgment section instead.

- Whenever this database is used for a study, the 'Terms of Use' clearly state that all versions of the database publications should be cited. This ensures that all data generators (as co-authors of the different database publications) also get credit for their contribution in enabling this database. The SISAL WG also encourages/and actively engages participation of all data contributors in subsequent research questions based on the database lead by different members of the WG, additionally upskilling and crediting original data contributors. See https://doi.org/10.1029/2021EO155315

I also have several additional comments about the data and code archived at
https://ora.ox.ac.uk/objects/uuid:ea1c56f7-384f-4df0-8525-6be00ab85870

- Please clarify if the code is final or if these scripts will be updated before publication. For example, the sisalv3_code is provided at the dataset link. This name implies that the scripts are new, but it is not clear that these scripts are any different from previous SISAL versions. For example, line 61 of the sisal_connect2db.R file is: "dbname='sisalv2',".
  - The sisalv3_codes were not designed to be exhaustive, and are rather added as WG members use the database for their own work and share their code. At times there are members using Julia and at other times members may be using R. In order to avoid confusion, in the README file we now specify that we provide codes that have been specifically modified for SISALv3 and add a note to say that additional codes are available for previous versions of the database and can be modified for SISALv3 as required by the users.  Thus, based on the work of current SISAL WG members, specific SISALv3 codes have been provided for Python, Julia and MySQL.

- Apologies if I missed this, but the README_SISALv3.pdf would be strengthened with an explanation of each zip folder provided. Specifically, it is not immediately clear to me what the sisalv3_standardised_trace_elements_datafiles.zip provides. My assumption is that these are the data measurements on their original depth (before downsampled)
- It is also not clear to me what the purpose of each file within the sisalv3_standardised_trace_elements_datafiles.zip is used for. I opened a random folder (sudwala) and looked at 2 excel sheets (trace_elements_datafile_sudwala_SC1_standardised.xlsx & trace_elements_datafile_sudwala_SC1.xlsx). As far as I can tell, these are the same data.
  - Thank you for pointing this out. We have expanded on the descriptions of the repository files included in the README file. We have added a sentence to the revised manuscript pointing interested readers toward the README file in the repository for descriptions of the datasets included.

- The authors provide several scripts which can be used to analyze the sql data in R/Python/etc. These include three R scripts (which I primary focused on for my review) which demonstrate how to load the data from MySQL, save the data as csv, and load the csv files. These scripts also provide the code for reproducing several figures from earlier SISAL publications. However, given that this is the 3rd version of the SISAL dataset, I am surprised to not see increased documentation about the usability and functionality of the dataset. There are two "low-hanging fruit" ways to improve the code documentation which immediately come to mind:
  - The first is to transition into using R Markdown / Jupyter notebooks. I find these resources extremely helpful when referencing code from other projects as I often want to see the output of the code before trying to reproduce the results

myself. Although this suggestion may seem trivial, the stated goal of SISAL to "make speleothem data to the wider paleoclimate community" necessities providing detailed documentation for accessing the data.

o The publishing of a new SISAL database version seems to provide an opportunity for create a central repository for code associated with analyzing SISAL data. This is especially relevant given the choice of SISAL to store the data in an SQL format. As at least one reviewer on a previous SISAL paper has noted, this is a data format that many (including me) are unaccustomed to (https://doi.org/10.5194/essd-2020-39-RC1). Therefore, providing more resources which demonstrate how to access and use the data would be valuable. Furthermore, there is now a long list of projects using previous versions of the SISAL database (as listed starting on line 90 of the main text) which could be referenced in this central repository.

- Increasing access to speleothem data is indeed one of our core goals. Towards this goal, we provide extensive documentation to access the database, and to mine the database using MySQL. In addition, a 'no-code' SISALwebApp has been created so that researchers who are not familiar with coding at all, especially primary data contributors, can also access the data. In addition, the webApp also provides MySQL code alongside the data download so that those who are keen to learn, can modify the code for more tailored analysis. In fact, the webApp has proven to be a useful tool for undergraduate paleoclimate courses.
- Majority of the publications that use the SISAL database are listed on the PAGES-SISAL webpage and can be easily found there.

**Referee Comment 3:**

**RC3: 'Comment on essd-2023-364', Anonymous Referee #3, 05 Nov 2023**

Citation: https://doi.org/10.5194/essd-2023-364-RC3

The present manuscript has documented the SISALv3 database, which includes new trace element data and more stable isotope records. The inclusion of trace element data adds value to the understanding of hydrological processes in the karst and cave. The authors also documented the revisions to the data structure. The updated database represents a community effort in synthesizing speleothem paleoclimate data with a global scope and thus is of great value to the community. The database can be used to benchmark climate model simulations and to study past changes in climate and hydrology.

The manuscript is very well written and fits nicely with the scope of Earth System Science Data. Below are a few comments that need to be addressed before the publication of this manuscript.

Thank you very much for your kind remarks and useful comments.

1. Since goal of the database is to allow "comparisons with isotope-enabled climate models and other earth system and hydrological models", it would be very helpful if the authors could provide example analysis code for this task. Specifically, the Last Glacial Maximum and the mid-Holocene have been the research focus of paleoclimate community for several decades (e.g., the Paleoclimate Modeling Intercomparison Project (PMIP); Kageyama et al., 2017, doi:10.5194/gmd-10-4035-2017; Otto-Bliesner et al., 2017, doi:10.5194/gmd-10-3979-2017). I would suggest the authors provide example codes (preferably using open-source program languages) for fetching stable isotope and/or trace element data for these key time periods and making simple plots. From the paleoclimate molders and proxy experts whom I have talked to, the SISAL database is not easy to use, in particular for model-data comparison. Providing example codes for PMIP users would greatly broaden the use of the database in paleoclimate research.

   - Thank you for your comment. It is good to hear that the database has proven useful for PMIP work. SISAL is an SQL database and since the first version of the database, we have been providing extensive documentation on how to access the database using MySQL, as well as coding examples to extract data covering specific time periods etc. These are available in the repository along with the database file. Over the last year, we have also developed the SISAL webApp which can be used for a 'no-code' approach to extracting data buy simply filtering data using site name, latitude-longitude bounds or age ranges (e.g. targeting the LGM or LIG). The SISAL webApp also provides MySQL code linked to the data search as a 'learning tool' which can be modified and used on platforms like MySQL Workbench for enhanced functionalities.
   - In addition, as you mention, there have been multiple publications on data-model comparisons using SISAL data and these publications provide additional information on screening criteria to use speleothem data for comparisons including the model benchmark periods. For example, see
     Comas-Bru L et al. (2019) Evaluating model outputs using integrated global speleothem records of climate change since the last glacial, Climate of the Past, 15, 1557-1579
     Parker SE, Harrison SP, Comas-Bru L, Kaushal N, LeGrande AN & Werner M (2021) A data–model approach to interpreting speleothem oxygen isotope records from monsoon regions, Climate of the Past, 17, 1119–1138

2. The database has undergone multiple version updates in the past few years. It would be interesting to briefly discuss the future plans from the working group. For example, is there plan for a new release in the future? Moreover, the author mentioned that the current database has included ~60%

of the known published data (Line 222), what are the challenges in including more dataset? Do the working group have a plan to cope with these challenges and increase the data inclusion?

- We have no more funding for further work on the database although we aim to apply. We hesitate to discuss future plans of the database in the manuscript before we have the funding in hand. Our experience with the database has shown that the first version has usually been able to add ~50% of the published proxy records (e.g. see Comas-Bru & Harrison, 2019). As the database is used for specific research questions, we have been able to target and add more data in subsequent versions. Since we have already started working on research questions based on trace element data, we can see that the same pattern is emerging with these proxy datasets as well.
- Comas-Bru L & Harrison SP, Eds: Harrison SP & Comas-Bru L (2019) SISAL: Bringing Added Value to Speleothem Research, Quaternary, 2(1), 7

3. Line 72: For clarification and comparison, please consider listing either the number of stable isotope and carbon isotope records in SISALv2 or the number of newly added records in SISALv3.

- Thank you for the suggestion. We have added this to **line section 80** of the revised manuscript: 'SISALv3, contains speleothem data from 365 sites from across the globe, including 95 Mg/Ca, 85 Sr/Ca, 52 Ba/Ca, 25 U/Ca, 29 P/Ca and 14 Sr-isotope records. The database also has increased spatiotemporal coverage for stable oxygen (892) and carbon (620) isotope records compared to SISALv2 (673 and 430 stable oxygen and carbon records, respectively).'